# Decentralized Projection-free Online Upper-Linearizable Optimization with Applications to DR-Submodular Optimization

**Yiyang Lu** *lu1202@purdue.edu*
*Purdue University, West Lafayette, IN, USA*
**Mohammad Pedramfar** *mohammad.pedramfar@mila.quebec*
*Mila - Quebec AI Institute/McGill University, Montreal, QC, Canada*
**Vaneet Aggarwal** *vaneet@purdue.edu*
*Purdue University, West Lafayette, IN, USA*

**Reviewed on OpenReview:** *https://openreview.net/forum?id=bZ5WD2HUQr*

## Abstract

We introduce a novel framework for decentralized projection-free optimization, extending projection-free methods to a broader class of upper-linearizable functions. Our approach leverages decentralized optimization techniques with the flexibility of upper-linearizable function frameworks, effectively generalizing optimization of traditional DR-submodular functions, which captures the 'diminishing return' property. We obtain the regret of $O(T^{1-\theta/2})$ with communication complexity of $O(T^\theta)$ and number of linear optimization oracle calls of $O(T^{2\theta})$ for decentralized upper-linearizable function optimization, for any $0 \leq \theta \leq 1$. This approach allows for the first results for monotone up-concave optimization with general convex constraints and non-monotone up-concave optimization with general convex constraints. Further, the above results for first order feedback are extended to zeroth order, semi-bandit, and bandit feedback.

## 1 Introduction

Modern machine learning systems, from multi-agent robotics to distributed sensor networks, increasingly rely on decentralized optimization to handle streaming data and objectives that evolve over time (Li et al., 2002; Xiao et al., 2007; Mokhtari et al., 2018). A prominent challenge in these systems is the online optimization of non-convex functions that exhibit a "diminishing returns (DR)" property, a characteristic formally captured by DR-submodularity. For instance, in power network reconfiguration (Mishra et al., 2017), distributed controllers across a smart grid must cooperatively adjust network switches to minimize system-wide power loss. This scenario is inherently decentralized: each controller uses local data, like power flow and voltage levels, to adjust its own switches, and communicates with its immediate neighbors to coordinate their actions toward the global goal of minimizing system-wide power loss. This communication involves exchanging proposed adjustments and local states, which are then aggregated using a weight matrix that determines the influence of each neighbor's input. This weight matrix is predetermined by the physical characteristics of the power network itself, such as the impedance of the power lines and the underlying network topology. The problem is also online because the optimal switch configuration must be continually updated to adapt to unpredictable shifts in consumer power demand and intermittent generation from renewable sources. The optimization task, which is to maximize power loss reduction, exhibits a diminishing returns property: the first few switch adjustments may yield substantial improvements, but the marginal benefit of each subsequent reconfiguration tends to decrease. This frames the problem as the maximization of a DR-submodular objective whose parameters change over time, making a decentralized, online solution essential. Such formulations also emerge in a wide array of other applications, including dynamic pricing, recommendation systems, inventory management, and mean-field variational inferences (Bian et al., 2019; Aldrighetti et al., 2021; Ito & Fujimaki, 2016; Hassani et al., 2017; Mitra et al., 2021; Gu et al., 2023). In all these scenarios, agents must cooperate to maximize a global objective, often with limited communication and time-varying objective functions. To

provide a unified solution for these scenarios, our work addresses $\gamma$-weakly up-concave functions, a class that generalizes both concave and DR-submodular functions.[1] The parameter $\gamma$ allows for a relaxation of this condition, with standard DR-submodular functions corresponding to the special case where $\gamma = 1$. We introduce these notions formally in Section 3.1.

Existing methods for online continuous DR-submodular optimization often fall into two broad categories: projection-based and projection-free. The first category, *projection-based strategies* (Chen et al., 2018b; Zhang et al., 2022), ensure that each decision remains within the feasible set by computing a Euclidean projection in each round, which requires solving quadratic program. The second category, *projection-free techniques* (Chen et al., 2018a;b; Zhang et al., 2019; Liao et al., 2023), replace projection operation with an efficient linear optimization oracle to update decisions. For many complex sets common in machine learning, the projection step is intractable for online settings, and the computational cost of an LOO is often orders of magnitude lower than that of a projection (Hazan, 2016; Braun et al., 2025). For instance, Braun et al. (2025) has shown in Table 1.1 that for problems constrained to a nuclear norm ball, projection requires a full singular value decomposition, whereas a linear program only requires finding the top singular vector pair, which is a much faster operation. Empirically, in our specific scenerio, Table 2 in Zhang et al. (2023) has shown that their projection-free algorithms are up to 6 times faster in practice than a projection-based counterpart for decentralized DR-submodular maximization, which motivate our paper's focus on developing a novel, projection-free framework.

However, in the decentralized realm, existing works in this area (Zhu et al., 2021; Zhang et al., 2023; Liao et al., 2023) often either achieve sub-optimal regret guarantees or require too much communication, and they are all restricted to narrow subclasses of functions, such as monotone 1-weakly up-concave (DR-submodular) functions over convex sets containing the origin, leaving a significant gap in the literature. Recently, the theoretical landscape was broadened by Pedramfar & Aggarwal (2024a), which introduced "*upper-linearizable functions*", a more general class that includes and extends DR-submodular and concave functions. This framework unifies optimization for various settings, including monotone and non-monotone up-concave functions over general convex sets along with monotone up-concave functions over convex sets containing the origin, and considers diverse feedback types. However, the analysis and algorithms for this powerful upper-linearizable optimization framework has so far been confined to centralized settings, where a single agent has access to all information.

This paper bridges that critical gap by introducing the first framework for **decentralized, online optimization of *upper-linearizable* functions**. Our primary objective is to develop **projection-free** algorithms that are not only computationally efficient by avoiding projections but are also tailored for the communication constraints inherent in decentralized networks. By extending the analysis from the centralized to the decentralized setting, we provide the first optimization results for several important problem classes, including online monotone and non-monotone up-concave maximization over general convex domains in the decentralized realm. For monotone DR submodular functions over convex sets containing the origin, with first order full-information feedback, which is the premise of prior works, this paper gives an algorithm that achieves a regret of $\mathcal{O}(T^{1-\theta/2})$ with a communication complexity of $\mathcal{O}(T^{\theta})$ and $\mathcal{O}(T^{2\theta})$ calls to a linear optimization oracle, where $\theta \in [0, 1]$ is a parameter that allows for an explicit trade-off between regret and communication overhead. Furthermore, our framework is versatile, enabling us to derive new results for various feedback models, including full-information, semi-bandit, and bandit settings, which were previously unexplored for decentralized DR-submodular optimization.

We also summarize our contributions and technical novelties of this work in details as follows.

1. **A General Framework for Decentralized Online Optimization of Upper-Linearizable Functions:** we introduce the first framework for decentralized, online optimization for the broad class of *upper-linearizable functions*. This function class is a generalization of up-concave (including DR-submodular and concave) functions, providing a significant leap over prior works that were restricted to monotone 1-weakly up-concave (i.e., DR-submodular) functions over convex set containing the origin under first-order full-information feedback (See Table 1). Our framework unifies the analysis of various settings, including

---

[1]It is well-known that any DR-submodular function is up-concave, i.e., concave along positive directions.

Table 1: Decentralized Online Up-concave Maximization Algorithms Comparison

| $F$ | Set | | Feedback | Reference | Appx. ($\alpha$) | $\log_T(\alpha$-Regret) | $\log_T$(Communication) | $\log_T$(LOO calls) | Range of $\theta$ |
|---|---|---|---|---|---|---|---|---|---|
| Monotone | $0 \in \mathcal{K}$ | $\nabla F$ | full information | DMFW (Zhu et al., 2021) | $1-e^{-1}$ | $1/2$ | $5/2$ | $5/2$ | - |
| | | | | Mono-DMFW (Zhang et al., 2023) | $1-e^{-1}$ | $4/5$ | $1$ | $1$ | - |
| | | | | DOBGA (Zhang et al., 2023) | $1-e^{-1}$ | $1/2$ | $1$ | $-$ | - |
| | | | | DPOBGA (Liao et al., 2023) | $1-e^{-1}$ | $3/4$ | $1/2$ | $1$ | - |
| | | | | Theorem 2 | $1-e^{-\gamma}$ | $1-\theta/2$ | $\theta$ | $2\theta$ | $[0, 1]$ |
| | | | semi-bandit | Theorem 4 | $1-e^{-\gamma}$ | $1-\theta/2$ | $\theta$ | $2\theta$ | $[0, 2/3]$ |
| | | $F$ | full information | Theorem 5 | $1-e^{-\gamma}$ | $1-\theta/4$ | $\theta$ | $2\theta$ | $[0, 1]$ |
| | | | bandit | Theorem 6 | $1-e^{-\gamma}$ | $1-\theta/4$ | $\theta$ | $2\theta$ | $[0, 4/5]$ |
| | general | $\nabla F$ | semi-bandit | Theorem 2 | $\gamma^2/(1+\gamma^2)$ | $1-\theta/2$ | $\theta$ | $2\theta$ | $[0, 1]$ |
| | | $F$ | bandit | Theorem 3 | $\gamma^2/(1+\gamma^2)$ | $1-\theta/4$ | $\theta$ | $2\theta$ | $[0, 1]$ |
| Non-Mono | general | $\nabla F$ | full information | Theorem 2 | $(1-p)/4$ | $1-\theta/2$ | $\theta$ | $2\theta$ | $[0, 1]$ |
| | | | semi-bandit | Theorem 4 | $(1-p)/4$ | $1-\theta/2$ | $\theta$ | $2\theta$ | $[0, 2/3]$ |
| | | $F$ | full information | Theorem 5 | $(1-p)/4$ | $1-\theta/4$ | $\theta$ | $2\theta$ | $[0, 1]$ |
| | | | bandit | Theorem 6 | $(1-p)/4$ | $1-\theta/4$ | $\theta$ | $2\theta$ | $[0, 4/5]$ |

Table 1 compares the results for different decentralized online up-concave maximization algorithms. $\nabla F$ refers to first-order query oracle while $F$ refers to zeroth-order query oracle. $\alpha$ refers to the approximation coefficient for the regret. The prior works only investigated monotone 1-weakly up-concave functions (i.e., only DR-submodular functions) over convex set containing the origin, while our results apply to $\gamma$-weakly up-concave functions for different scenarios, hence the differences in $\alpha$. Here $p := \min_{\mathbf{z} \in \mathcal{K}} \|\mathbf{z}\|_\infty$. Communication refers to the total number of communications, and LOO calls refers to the total number of calls to the Linear Optimization Oracle. The results hold for any $\theta$ in the range specified in the last column. Note that DOBGA is projection-based and requires prjection oracle, while all others are projection-free and utilize linear optimization oracle.

Notably, all prior works are confined to a very narrow subclass of functions: monotone 1-weakly up-concave (i.e., DR-submodular) functions over convex sets containing the origin. In contrast, our framework provides the first guarantees for a much broader range of problems, including general $\gamma$-weakly, non-monotone functions, and optimization over general convex domains. It also introduces a flexible trade-off between regret and communication via the parameter $\theta$. Even when specialized, our results are highly competitive: setting $\theta = \frac{1}{2}$ matches the state-of-the-art projection-free method (DPOBGA), while $\theta = 1$ matches the regret of the best projection-based method (DOBGA).

monotone and non-monotone $\gamma$-weakly up-concave functions over various convex sets – by treating them as specific instances of the upper-linearizable class.

2. **An Efficient Algorithm for the General Framework with a Principled Trade-off:** In section 4, we propose a single, versatile algorithm, `DROCULO` (Alg. 1), for the general class of upper-linearizable functions. The algorithm operates with either semi-bandit or first-order full-information feedback, depending on the structure of the specific function class being optimized. It achieves an explicit trade-off between statistical performance and resource efficiency: for any parameter $\theta \in [0, 1]$, it attains a regret of $O(T^{1-\theta/2})$ with a communication complexity of $O(T^\theta)$ (Theorem 2). Even when applied to up-concave maximization, this provides the first results for several general classes, including monotone and non-monotone $\gamma$-weakly up-concave functions over various convex domains.

3. **Extension to Different Feedback for Up-Concave Subclasses:** In section 5, we demonstrate the practical utility of our framework by applying it to prominent up-concave subclasses. This specialization allows us to develop the first algorithms for these classes under restrictive feedback settings (Alg. 2-5). By extending meta-algorithmic techniques by (Pedramfar & Aggarwal, 2024a) from centralized to decentralized setting, we design specialized solutions for:

   - monotone $\gamma$-weakly up-concave functions over general convex sets (Case B.1): We provide Alg. 2 for the restrictive *bandit* feedback setting.
   - monotone up-concave functions over convex sets containing the origin (Cases B.2) & non-monotone up-concave functions over general convex sets (Cases B.3): We introduce a series of novel extensions for *semi-bandit* (Alg. 3), *zeroth-order full-information* (Alg. 4), and *bandit* feedback (Alg. 5).
   - In total, our framework yields 10 algorithms for up-concave maximization (two for Case B.1, four for Case B.2, and four for Case B.3), 9 of which are the first results in their respective feedback settings.

**Technical Novelty**

1. We note that previous works on decentralized DR-submodular optimization assumed monotone 1-weakly DR-submodular functions with $0 \in \mathcal{K}$. Thus, we needed a non-trivial approach to extend the setup to include non-monotonicity of the function, $\gamma$-weakly DR-submodular functions, and general convex set constraints. This is done using the notion of upper-linearizable functions, allowing us to obtain the first results for (i) monotone $\gamma$-weakly up-concave functions with general convex sets, (ii) monotone $\gamma$-weakly up-concave functions over convex sets with $0 \in \mathcal{K}$, and (iii) non-monotone up-concave functions with general convex sets.

2. We proved that the framework proposed by Pedramfar & Aggarwal (2024a) which was proposed for and examined in the centralized setting, can be extended to the decentralized setting with proper adaptations. The notion of regret changes in the decentralized setting, where we have to consider the average of loss function among all agents instead of just one function, as is the case in a the centralized setting. Note that the changes in the definition of online optimization between centralized and decentralized optimization makes applications of meta-algorithms used in Pedramfar & Aggarwal (2024a) non-trivial. The centralized version have no notion of communication between nodes and it requires nuance in how one goes about applying meta-algorithms designed for centralized setting to base algorithms that are decentralized.

3. In the cases of monotone functions over convex sets containing the origin and non-monotone functions, the main algorithm, i.e., `DDROCULO`, requires first order full-information feedback. In the centralized setting, the `SFTT` meta-algorithm, described in Pedramfar & Aggarwal (2024a), is designed to convert the such algorithms into algorithms only requiring semi-bandit feedback. However, this algorithm does not directly work in the decentralized setup. We design Algorithms 3 and 5 using the idea of `SFTT` meta-algorithm. The challenge here is to ensure that the `SFTT` blocking mechanism interacts properly with the existing blocking mechanism of the base algorithm.

The remainder of this paper is organized as follows. In Section 2, we review related work. In Section 3, we establish the necessary preliminaries, including our notation, problem formulation, and key definitions for upper-linearizable functions. In Section 4, we present our main algorithm, `DROCULO`, with its theoretical guarantees for the general upper-linearizable class. In Section 5, we extend this framework by developing specialized algorithms for prominent subclasses of up-concave functions under various feedback settings, including semi-bandit and bandit feedback. All detailed proofs are deferred to the appendices.

## 2 Related Works

**Decentralized Online Convex Optimization:** Zinkevich et al. (2010) presented a decentralized primal-dual gradient method using local communication and dual averaging over static networks, and achieved $\tilde{O}(\sqrt{T})$ regret for Lipschitz losses, yet it assumed static connectivity and full gradient access, limiting adaptability in dynamic settings. Yan et al. (2012) extended this via distributed projected gradient descent across agents with standard communication steps, proving regret bounds $O(n^{5/4}\rho^{-1/2}\sqrt{T})$ for convex functions, but it suffered from polynomial dependence on $n$ and spectral gap $\rho$. Lee et al. (2016) proposed ODA-C and ODA-PS, incorporating Nesterov-style dual-averaging on static and dynamic graphs, achieving similar $O(\sqrt{T})$ regret; still, its performance degrades in heterogeneous or rapidly changing networks. Wan et al. (2024) introduced AD-FTGL with online accelerated communication, tightening regret to $\tilde{O}(n\,\rho^{-1/4}\sqrt{T})$ for convex losses, and proved matching lower bounds—closing gaps in earlier algorithms' dependence on $\rho$ and $n$.

**Upper-Linearizable Function and its Online Algorithm:** In late 2024, Pedramfar & Aggarwal (2024a) introduced the concept of upper-linearizable functions, a class that generalizes up-concavity (concavity and DR-submodularity) across various settings, including both monotone and non-monotone cases over different convex sets. They also explored projection-free algorithms in *centralized* settings for these setups. Additionally, they proposed several meta-algorithms that adapt the feedback type, converting full-information queries to trivial queries and transitioning from first-order to zeroth-order feedback. These ideas were further generalized in (Pedramfar et al., 2025) by using "uniform wrappers"; a framework to convert algorithms for online concave optimization to online upper-linearizable optimization (in the centralized setting). Pedramfar & Aggarwal (2024a) demonstrated the generality of this class of functions by showing that it includes (i) monotone $\gamma$-weakly up-concave functions over general convex sets (Case B.1), (ii) monotone $\gamma$-weakly up-concave functions over

convex sets containing the origin (Case B.2), and (iii) non-monotone up-concave optimization over general convex sets (Case B.3). The details of these functions are provided in Appendix B for completeness, and Lemma 4, 5, 7 demonstrate how these prominent up-concave functions are upper-linearizable. However, the framework of 'Upper-Linearizable functions' is not restricted to these three examples. While these functions are central to contemporary research in submodular optimization, the 'Upper-Linearizable' concept is relatively new, and its full scope is a subject for future exploration.

**Decentralized Online DR-Submodular Maximization:** Prior works in decentralized online DR-submodular maximization have largely followed two algorithmic paths. One path, rooted in the Frank-Wolfe method, was initiated by the Decentralized Meta-Frank-Wolfe (DMFW) algorithm from Zhu et al. (2021), which achieved an $O(\sqrt{T})$ regret but at a high communication cost of $O(T^{5/2})$. This limitation was later addressed by by the Mono-DMFW from Zhang et al. (2023), which improved regret to $O(T^{4/5})$ while reducing communication complexity to $O(T)$. The other path leverages boosting gradient ascent, introduced with the projection-based DOBGA algorithm from Zhang et al. (2023), which obtained an $O(T^{1/2})$ regret with $O(T)$ communication. The principles of this method were subsequently adapted to a projection-free context by Liao et al. (2023) in their DPOBGA algorith, reporting $O(T^{3/4})$ regret with $O(T^{1/2})$ communication.

Notably, these prior works were confined to maximizing monotone DR-submodular functions over convex sets containing the origin. This setting, corresponding to 1-weakly up-concave functions, restricts all prior methods to the same $1 - 1/e$ approximation ratio, and represents a special instance of the general problem classes we address (Appendix B.2).

In this paper, we present algorithms for optimizing upper-linearizable functions, achieving a regret bound of $O(T^{1-\theta/2})$, in first order feedback case, and $O(T^{1-\theta/4})$, in zeroth order feedback case, with a communication complexity of $O(T^\theta)$ and number of LOO calls of $O(T^{2\theta})$. In first order full-information case, for $\theta = 1$, we show that the regret and the communication complexity matches the best known projection-based algorithm in Zhang et al. (2023) and for $\theta = 1/2$, the results match that in Liao et al. (2023) in the special case of monotone 1-weakly DR-submodular functions with the convex set containing the origin.

## 3 Preliminaries

### 3.1 Notations and Definitions

This paper considers a decentralized setting involving $N$ agents connected over a network represented by an undirected graph $G = (V, E)$, where $V = \{1, \cdots, N\}$ is the set of nodes, and $E \subseteq V \times V$ is the set of edges. Each agent $i \in V$ acts as a local decision maker and can communicate only with its neighbors, defined as $\mathcal{N}_i = \{j \in V \mid (i, j) \in E\} \cup \{i\}$. To model the communication between agents, we introduce a non-negative weight matrix $A = [a_{i,j}] \in \mathbb{R}_+^{N \times N}$, which is supported on the graph $G$. The matrix $A$ is symmetric and doubly stochastic, with $a_{i,j} > 0$ only if $(i, j) \in E$ or $i = j$. Since $A^\top = A$ and $A\mathbf{1} = \mathbf{1}$, where $\mathbf{1}$ denotes the vector of all ones, 1 is the largest eigenvalue of $A$ with $\mathbf{1}$ being the eigenvector. When agent $i$ *communicates* with its neighbors $\mathcal{N}_i$, it exchanges a local state vector $\mathbf{s}^i$ based on the weight matrix $A = \{a_{ij}\}$. In other words, each agent $i$ receives the weighted average of the local states of its neighbors, i.e., $\sum_{j \in \mathcal{N}_i} a_{ij}\mathbf{s}^j$. It is common to assess decentralized optimization algorithms using the total number of communications with respect to total rounds $T$, which is denoted as "communication" in Table 1. These are formally captured in Assumption 1.

**Assumption 1.** *We assume that the communication weight matrix $A$ is symmetric and doubly stochastic, whose second largest eigenvalue, $\lambda_2$, is strictly less than 1.*

*Remark* 1. As mentioned, $A$ dictates how information is shared and averaged across the network, and its largest eigenvalue is always $\mathbf{1}$. The rate of convergence, however, is captured by the second-largest eigenvalue, $\lambda_2$, which reflects the network's connectivity. A smaller $\lambda_2$ implies a better-connected network and faster convergence. Assumption 1 is standard in decentralized optimization literature (Zhu et al., 2021; Zhang et al., 2023; Liao et al., 2023). If it were violated (i.e. $\lambda_2 = 1$), the network would be disconnected, composed of at least two isolated sub-groups of agents that cannot communicate with each other. In such a scenario, a global agreement is impossible, and the algorithm would fail to find a solution for the network-wide objective.

A set $\mathcal{K} \subseteq \mathbb{R}^d$ is convex if $\forall \mathbf{x}, \mathbf{y} \in \mathcal{K}$ and $\forall \alpha \in [0, 1]$, we have $\alpha \mathbf{x} + (1 - \alpha)\mathbf{y} \in \mathcal{K}$. For a constrained set $\mathcal{K}$, we denote the radius of the set as $R \triangleq \max \|\mathbf{x}\|, \forall \mathbf{x} \in \mathcal{K}$. For two vectors $\mathbf{x}, \mathbf{y} \in \mathcal{K}$, we say $\mathbf{x} \leq \mathbf{y}$ if every element in $\mathbf{x}$ is less than or equal to the corresponding element in $\mathbf{y}$. Given a set $\mathcal{K}$, we define a function class $\mathcal{F}$ as a subset of all real-valued functions over $\mathcal{K}$. For a set $\mathcal{K} \subseteq \mathbb{R}^d$, we define its *affine hull* $\mathrm{aff}(\mathcal{K})$ to be the set of $\alpha \mathbf{x} + (1 - \alpha)\mathbf{y}$ for all $\mathbf{x}, \mathbf{y} \in \mathcal{K}$ and $\alpha \in \mathbb{R}$. The *relative interior* of $\mathcal{K}$ is defined as $\mathrm{relint}(\mathcal{K}) := \{\mathbf{x} \in \mathcal{K} \mid \exists r > 0, \mathbb{B}_r(\mathbf{x}) \cap \mathrm{aff}(\mathcal{K}) \subseteq \mathcal{K}\}$. Assumptions of the feasible set $\mathcal{K}$ [2] is given as follows.

**Assumption 2.** *We assume the feasible set $\mathcal{K}$ is convex and compact with radius $R$, i.e., $R = \max_{\mathbf{x} \in \mathcal{K}} \|\mathbf{x}\|$.*

Given $0 < \gamma \leq 1$, we say a differentiable function $f : \mathcal{K} \to \mathbb{R}$ is $\gamma$-*weakly up-concave* if it is $\gamma$-weakly concave along positive directions. Specifically if, $\forall \mathbf{x} \leq \mathbf{y} \in \mathcal{K}$, we have

$$\gamma \left( \langle \nabla f(\mathbf{y}), \mathbf{y} - \mathbf{x} \rangle \right) \leq f(\mathbf{y}) - f(\mathbf{x}) \leq \frac{1}{\gamma} \left( \langle \nabla f(\mathbf{x}), \mathbf{y} - \mathbf{x} \rangle \right). \tag{1}$$

A differentiable function $f : \mathcal{K} \to \mathbb{R}$ is called $\gamma$-*weakly continuous DR-submodular* if $\forall \mathbf{x} \leq \mathbf{y}$, we have $\nabla f(\mathbf{x}) \geq \gamma \nabla f(\mathbf{y})$. It follows that any $\gamma$-weakly continuous DR-submodular functions is $\gamma$-weakly up-concave. Note that DR-submodular functions are a special case of up-concave functions, though the term "DR-submodular" is more common. For generality, this paper will use the broader "up-concave" term, as our results for this class apply directly to DR-submodular functions. In subsequent sections, we will demonstrate that these up-concave functions are prominent examples of the broader upper-linearizable class.

Given a DR-submodular function, even the maximization problem in the centralized offline setup is NP-hard. For example, when $f$ is a monotone continuous DR-submodular function and $\mathcal{K} \subseteq [0, 1]^d$ contains the origin, finding a point $\mathbf{x} \in \mathcal{K}$ such that $f(\mathbf{x})$ is optimal is NP-hard. More generally, for any $\epsilon > 0$, finding any point $\mathbf{x} \in \mathcal{K}$ such that $f(\mathbf{x})$ is at least $(1 - e^{-1} + \epsilon)$ times the optimal value is NP-hard. (See Proposition 3 in Bian et al. (2017)). However, there are polynomial times algorithms that achieve the ratio of $1 - e^{-1}$. The ratio with this property is referred to as the *optimal approximation ratio*, in this case being $1 - e^{-1}$. In the three settings considered in this paper, (in the special case of $\gamma = 1$) the approximation ratios for monotone function over convex sets containing the origin and non-monotone functions over general convex sets are known to be optimal.(See Bian et al. (2017); Mualem & Feldman (2023)) In the case of monotone functions over general convex sets (when $\gamma = 1$), the approximation ratio of $1/2$ is the best known in the literature so far and it is conjectured to be optimal (See Pedramfar et al. (2023)).

Given a function $f : \mathcal{K} \to \mathbb{R}$, we define a query oracle $\mathcal{Q}$ for $f$ to be a function on $\mathcal{K}$ that takes a point of query $\mathbf{w}$, and returns a noisy response $\mathbf{o}$. We say $\mathcal{Q}$ is a *first order* oracle if $\forall \mathbf{w} \in \mathcal{K}$, $\mathcal{Q}(\mathbf{w})$ is a random vector in $\mathbb{R}^d$ such that $\mathbb{E}[\mathcal{Q}(\mathbf{w})] = \nabla f(\mathbf{w})$. We say $\mathcal{Q}$ is a *zeroth order* oracle if $\forall \mathbf{w} \in \mathcal{K}$, $\mathcal{Q}(\mathbf{w})$ is a random variable in $\mathbb{R}$ such that $\mathbb{E}[\mathcal{Q}(\mathbf{w})] = f(\mathbf{w})$. The role of $\mathcal{Q}$ is to provide us information about the function $f$; in other words, the only way we obtain information about $f$ is by querying $\mathcal{Q}$.

### 3.2 Upper-Linearizable Functions

Recently, Pedramfar & Aggarwal (2024a) proposed the notion of upper-linearizable function, which generalizes the notion of up-concave and DR-submodular. The function class $\mathcal{F}$ is upper-linearizable if there exists $\mathfrak{g} : \mathcal{F} \times \mathcal{K} \to \mathbb{R}^d$, $h : \mathcal{K} \to \mathcal{K}$, and constant $0 < \alpha \leq 1$ and $\beta > 0$ such that $\forall \mathbf{x}, \mathbf{y} \in \mathcal{K}, \forall f \in \mathcal{F}$:

$$\alpha f(\mathbf{y}) - f(h(\mathbf{x})) \leq \beta \left( \langle \mathfrak{g}(f, \mathbf{x}), \mathbf{y} - \mathbf{x} \rangle \right). \tag{2}$$

As mentioned in the Section 2 with details in Appendix B, Pedramfar & Aggarwal (2024a) has demonstrated that three common objectives in the submodular optimization communities are upper-linearizable:

   (i) monotone $\gamma$-weakly up-concave functions with general convex sets (Case B.1),

  (ii) monotone $\gamma$-weakly up-concave functions over convex sets with $0 \in \mathcal{K}$ (Case B.2), and

 (iii) non-monotone up-concave functions with general convex sets (Case B.3).

---

[2]In our setting, at each round, every agent $i$ selects a decision from a common feasible set $\mathcal{K} \subseteq \mathbb{R}^d$, which we refer to as the action set. The detailed problem setting is given below in Section 3.3.

In Appendix B, we also provide the corresponding $\alpha$, $\beta$, $h(\cdot)$ and $\mathfrak{g}(\cdot)$ mapping, and their linearizable query oracles. As we will see later on, given an upper-linearizable function class $\mathcal{F}$, the constant $\alpha$ corresponds to the approximation ratio of our algorithms when the objective functions are in $\mathcal{F}$; specifically, it serves as the approximation coefficient for the regret.

When optimizing upper-linearizable functions, we assume that an oracle $\mathcal{G}$ is provided, which we call *linearizable query oracle*, that returns an unbiased estimate of $\mathfrak{g}(f, \mathbf{x})$. We refer to Appendix B for examples of such oracles for the three cases of upper-linearizable functions.

We note that the function $h$ allows us to consider more general functions. For example, $h(\cdot)$ takes identity function for case B.1 and case B.2, while $h(\mathbf{x}) = \frac{\mathbf{x} + \bar{\mathbf{x}}}{2}$ for some constant $\bar{\mathbf{x}} \in \mathcal{K}$ for case B.3. A detailed discussion on the role of the function $h$ is presented in Appendix A.

As mentioned, the framework of 'upper-linearizable functions' is not limited to these examples, although they are the most well-known function classes to submodular optimization community. The 'upper-linearizable' framework is a recent development, and it is anticipated that further applications will be identified as the concept matures.

### 3.3 Problem Formulation

In a decentralized setting, given a function class $\mathcal{F}$, the adversary chooses a sequence of objective functions $f_{t,i} \in \mathcal{F}$ and the corresponding query oracles $\mathcal{Q}_{t,i}$, for round $t \in [T]$ and agent $i \in [N]$. Note that the query oracle is the only way the agent can get any information on the objective function. In $t^{th}$ round, agent $i$ selects a pair of points $\hat{\mathbf{x}}_t^i, \mathbf{w}_t^i \in \mathcal{K}$, plays $\hat{\mathbf{x}}_t^i$, queries the provided oracle at $\mathbf{w}_t^i$, and observes oracle response $\mathbf{o}_t^i = \mathcal{Q}_{t,i}(\mathbf{w_t^i})$. [3]. The agent $i$ may communicate a local state vector $\mathbf{s}_t^i$ with its neighbors $\mathcal{N}_i$ as per Section 3.1, i.e., each agent $i$ receives the weighted average of the local states of its neighbors $\sum_{j \in \mathcal{N}_i} a_{ij} \mathbf{s}_t^j$.

We say the queries are *trivial* if $\mathbf{w}_t^i = \hat{\mathbf{x}}_t^i$, otherwise we say they are *non-trivial*. Note that in the three up-concave functions we mention in Related Works with details in Appendix B, case B.1 has trivial queries while case B.2 and B.3 have non-trivial queries, with Algorithm 7 determining the point of query for case B.2 and Algorithm 8 for case B.3, respectively.

The goal for each agent $i$ is to optimize the aggregate function over the network over time: $\sum_{t=1}^T \sum_{j=1}^N f_{t,j}(\mathbf{x}_t^i)$ (Zhu et al., 2021; Zhang et al., 2023). For any approximation coefficient $0 < \alpha \leq 1$, we define the $\alpha$-*regret* for agent $i$ to be:

$$\mathcal{R}_\alpha^i := \alpha \max_{\mathbf{u} \in \mathcal{K}} \frac{1}{N} \sum_{t=1}^T \sum_{j=1}^N f_{t,j}(\mathbf{u}) - \frac{1}{N} \sum_{t=1}^T \sum_{j=1}^N f_{t,j}(\hat{\mathbf{x}}_t^i).$$

*Remark* 2. As per our earlier discussion in Section 3.1, if we set $\alpha = 1$, obtaining a sublinear $\alpha$-regret even the offline centralized version of the problem could be NP-hard. Thus, the goal is to find the highest $\alpha$ possible and minimize the $\alpha$-regret for such a choice of $\alpha$. *As shown in Pedramfar & Aggarwal (2024a), if a function class is upper-linearizable with constant $\alpha$ (as per Equation 2), then there are algorithms obtaining sub-linear $\alpha$-regret in the corresponding offline (and online) optimization problems.* In Cases B.2 and B.3, (in the case $\gamma = 1$) the optimal approximation coefficient for the offline problem is known (See Bian et al. (2017); Mualem & Feldman (2023)) and the function classes in B.2 and B.3 are upper-linearizable with these approximation coefficients. Moreover, for case B.1, the function class is linearizable with the coefficient $\gamma^2/(1 + c\gamma^2)$ which is the best known approximation coefficient for the corresponding offline optimization problem. [4] Thus, among the results in this work for Cases B.1-B.3, it is only for the case B.1 where a higher approximation coefficient is not yet theoretically ruled out.

We say the agent takes *semi-bandit feedback* if the adversary provides first-order oracle and the agents have trivial queries. More formally, the query oracle returns a noisy response with mean of $\nabla f_{t,i}(\hat{\mathbf{x}}_t^i)$. Similarly,

---

[3]In general, the agent may query more than one point; in other words, it selects an action point $\mathbf{x}_t^i$ and a sequence of points $(\mathbf{w}_t^i)_j$ for $j \in [k]$ for some $k \geq 1$ which may depend on $t$ and $i$. However, in all algorithms considered in this paper, agents only require a single query.

[4]In fact, as mentioned earlier, at least in the case $c = \gamma = 1$, this coefficient is conjectured to be optimal. (See Pedramfar et al. (2023))

we say the agent takes *bandit feedback* if the oracle is zeroth-order and the agents have trivial queries, i.e., the query oracle returns a noisy response with mean of $f_{t,i}(\hat{\mathbf{x}}_t^i)$. If any agent has non-trivial queries, we say the agent requires *full-information feedback*. More formally, first-order full-information feedback returns a noisy response with mean of $\nabla f_{t,i}(\mathbf{w}_t^i)$ and similar for zeroth-order, $f_{t,i}(\mathbf{w}_t^i)$.

### 3.4 Set Oracles

The action set is a given convex set $\mathcal{K}$. However, the way such a set is given could be quite complicated. For example, it could be given as intersection of hyperplanes, or balls, or by some more complicated equations. Naturally, obtaining information about $\mathcal{K}$ depends on the way it is characterized. In order to abstract away this complexity, the notion of *set oracle* is defined. *Besides some general information given at the beginning, the only way the algorithm may obtain information about $\mathcal{K}$ is through such a set oracle.* The most common set oracles considered in the literature are *linear optimization oracle (LOO)* and *projection oracle*. As mentioned in Section 1, projection oracles could be computationally costly since such an oracle minimizes Euclidean distance, which is a quadratic optimization problem. To avoid problems caused by projection oracle, we use an *infeasible projection oracle* which is implemented using an LOO, and we introduce these concepts formally in the following.

A **projection oracle** $\mathcal{O}_P$ takes any point $\mathbf{x} \in \mathbb{R}^d$ as input and returns the unique point in $\mathcal{K}$ that is closest to $\mathbf{x}$ with respect to the Euclidean norm, formally computing $\mathcal{O}_P(\mathcal{K}, \mathbf{x}) := \arg\min_{\mathbf{u} \in \mathcal{K}} \|\mathbf{x} - \mathbf{u}\|_2$. This oracle is central to projection-based algorithms, which ensure feasibility by projecting iterates back onto the set $\mathcal{K}$. A **linear optimization oracle** ($\mathcal{O}_{LO}$) takes a linear objective, defined by a vector $\mathbf{x} \in \mathbb{R}^d$, as input and returns a point in $\mathcal{K}$ that maximizes this objective. In other words, for any given $\mathbf{x} \in \mathbb{R}^d$, the oracle solves the problem $\mathcal{O}_{LO}(\mathcal{K}, \mathbf{x}) := \arg\max_{\mathbf{u} \in \mathcal{K}} \langle \mathbf{x}, \mathbf{u} \rangle$. This oracle is the core of projection-free methods, which avoid costly quadratic projection problems by instead solving a sequence of linear problems over the feasible set.

Given a set $\mathcal{K} \subseteq \mathbb{R}^d$, we define a point $\tilde{\mathbf{y}} \in \mathbb{R}^d$ to be the infeasible projection of $\mathbf{y} \in \mathbb{R}^d$ onto set $\mathcal{K}$, if $\forall \mathbf{z} \in \mathcal{K}$ we have $\|\tilde{\mathbf{y}} - \mathbf{z}\|^2 \le \|\mathbf{y} - \mathbf{z}\|^2$. Given a set $\mathcal{K}$ and an error tolerance parameter $\epsilon$, we define an **infeasible projection oracle** $\mathcal{O}_{IP}$ to be an algorithm that takes a pair of points $\mathbf{x} \in \mathcal{K}$ and $\mathbf{y} \in \mathbb{R}^d$, returns a pair of points $\mathbf{x}' \in \mathcal{K}$ and $\tilde{\mathbf{y}} \in \mathbb{R}^d$, where $\mathbf{y}'$ is an infeasible projection of $\mathbf{y}$ unto set $\mathcal{K}$, and $\|\mathbf{x}' - \tilde{\mathbf{y}}\|^2 \le 3\epsilon$. Specifically, we look at infeasible projection oracles implemented through linear optimization oracles. We also introduce a useful lemma derived from (Liao et al., 2023) for the infeasible projection oracles.

**Lemma 1** (Lemma 1 in Liao et al. (2023)). *Let $\mathcal{B}$ be the unit ball centered at the origin. There exists an algorithm $\mathcal{O}_{IP}$ referred to as infeasible projection oracle over any convex set $\mathcal{K} \subseteq R\mathcal{B}$ (where $R\mathcal{B}$ means a ball with radius $R$), which takes the set $\mathcal{K}$, a pair of points $(\mathbf{x}_0, \mathbf{y}_0) \in \mathcal{K} \times \mathbb{R}^n$, and an error tolerance parameter $\epsilon$ as the input, and can output*

$$(\mathbf{x}, \tilde{\mathbf{y}}) = \mathcal{O}_{IP}(\mathbf{x}_0, \mathbf{y}_0, \epsilon)$$

*such that $(\mathbf{x}, \tilde{\mathbf{y}}) \in \mathcal{K} \times R\mathcal{B}$, $\|\mathbf{x} - \tilde{\mathbf{y}}\|^2 \le 3\epsilon$, and $\forall \mathbf{z} \in \mathcal{K}$, $\|\tilde{\mathbf{y}} - \mathbf{z}\|^2 \le \|\mathbf{y}_0 - \mathbf{z}\|^2$. Moreover, such an oracle $\mathcal{O}_{IP}$ can be implemented with total LOO calls bounded by*

$$\left\lceil \frac{27R^2}{\epsilon} - 2 \right\rceil \max\left(1, \frac{\|\mathbf{x}_0 - \mathbf{y}_0\|^2(\|\mathbf{x}_0 - \mathbf{y}_0\|^2 - \epsilon)}{4\epsilon^2} + 1\right). \tag{3}$$

## 4 Main Result for Decentralized Online Upper-Linearizable Optimization

In this section, we introduce our main algorithm, DecentRalized Online Continuous Upper-Linearizable Optimization (`DROCULO`, like Dracula), which is a *projection-free* method designed for the broad class of upper-linearizable functions. The theoretical analysis that establishes the performance guarantees of our method relies on the following standard assumptions. Note that these assumptions apply specifically to the general analysis in this section; assumptions for the specialized cases in Section 5 will be introduced therein.

**Assumption 3.** *All objective functions $f_{t,i} \in \mathcal{F} : \mathcal{K} \to \mathbb{R}$ are $M_1$-Lipschitz continuous, differentiable, and upper-linearizable with $\alpha, \beta, \mathfrak{g}$ and $h$ as defined in Equation 2. We also assume that $\mathfrak{g}(f, \mathbf{x})$ is $L_1$-Lipschitz with respect to the second term $\mathbf{x}$.*

Note that it is common assumption in the literature (Liao et al., 2023; Fazel & Sadeghi, 2023; Zhang et al., 2022; 2024) for the underlying up-concave function to be smooth. In the three examples of upper-linearizable functions we considered, the smoothness of the underlying up-concave function implies Assumption 3 (See Lemma 6, Lemma 8).

**Assumption 4.** *Given the upper-linearizable function class $\mathcal{F}$, The linearizable query oracle $\mathcal{G}$ access each $f_{t,i} \in \mathcal{F}$ through a first-order query oracle $\mathcal{Q}$ such that its response for an input $\mathbf{x}_t^i$ is an unbiased estimate of $\mathfrak{g}(f_{t,i}, \mathbf{x}_t^i)$, i.e., $\mathbb{E}[\mathcal{G}(\mathbf{x}_t^i)] = \mathfrak{g}(f_{t,i}, \mathbf{x}_t^i)$. We further assume that the responses $\mathbf{o}_t^i$ of the linearizable query oracle $\mathcal{G}$ are bounded by $G$, i.e., $\|\mathbf{o}_t^i\| = \|\mathcal{G}(\mathbf{x}_t^i)\| \leq G$.*

Note that Assumption 4 holds for the three examples of upper-linearizable functions detailed in Appendix B (see Algorithms 7 and 8). The linearizable query oracle $\mathcal{G}$ serves as an abstraction layer. For a given input $\mathbf{x}_t^i$, $\mathcal{G}$ determines the appropriate point $\mathbf{w}_t^i$ to query the underlying first-order oracle $\mathcal{Q}$ to produce the required unbiased estimate of $\mathfrak{g}(f_{t,i}, \mathbf{x}_t^i)$. Thus, Assumption 4 is satisfied for these cases if the underlying first-order query oracle is bounded.

## 4.1 Algorithm

The `DROCULO` algorithm is presented in Algorithm 1. Without loss of generality we assume that $T \bmod K = 0$. It operates by partitioning the time horizon $T$ into $T/K$ blocks, namely $\mathcal{T}_m = \{(m-1)K+1, \ldots, mK\}$ for block $1 \leq m \leq T/K$, where $K$ is the block size. Each agent $i$ maintains a local state vector $\mathbf{s}_m^i$ constituted of a decision variable $\mathbf{x}_m^i \in \mathcal{K}$ and an auxiliary variable $\tilde{\mathbf{y}}_m^i$ for the infeasible projection, which are initialized to a common point $\mathbf{c} \in \mathcal{K}$ (line 2).

The algorithm proceeds in blocks (line 3). For each block $m$, all agents perform the following steps in parallel. At each time step $t$ within block $m$, agent $i$ plays the action $\hat{\mathbf{x}}_t^i = h(\mathbf{x}_m^i)$ (line 6), where $h(\cdot)$ is the transformation map associated with the upper-linearizable function class. Note that the action played, $\hat{\mathbf{x}}_t^i$, may differ from the agent's internal decision variable, $\mathbf{x}_m^i$. The agent then queries the linearizable oracle $\mathcal{G}$ at $\mathbf{x}_m^i$ to obtain a response $\mathbf{o}_t^i$ (line 7). The specific query point $\mathbf{w}_t^i$ is handled internally by the oracle $\mathcal{G}$ to produce an unbiased estimate of $\mathfrak{g}(f_{t,i}, \mathbf{x}_m^i)$.

At the end of each block, agent $i$ communicates its local state $\mathbf{s}_m^i = (\mathbf{x}_m^i, \tilde{\mathbf{y}}_m^i)$ with its immediate neighbors $\mathcal{N}_i$, i.e., it receives an aggregated state vector $(\sum_{j \in \mathcal{N}_i} a_{ij} \mathbf{x}_m^j, \sum_{j \in \mathcal{N}_i} a_{ij} \tilde{\mathbf{y}}_m^j)$ (line 9). This information is used to compute an intermediate variable $\mathbf{y}_{m+1}^i$ (line 10), which incorporates a scaled average $\eta \sum_{t \in \mathcal{T}_m} \mathbf{o}_t^i = \eta K \left( \frac{1}{K} \sum_{t \in \mathcal{T}_m} \mathbf{o}_t^i \right)$ of the oracle responses from the block. Note that this averaging is the main purpose of the blocking mechanism as it allows us to reduce the variance of the estimates obtained from the query oracle. Finally, the agent updates its decision and auxiliary variables for the next block, $(\mathbf{x}_{m+1}^i, \tilde{\mathbf{y}}_{m+1}^i)$, by performing an infeasible projection using the oracle $\mathcal{O}_{IP}$ with an error tolerance $\epsilon$ (line 11).

---

**Algorithm 1** DecentRalized Online Continuous Upper-Linearizable Optimization - `DROCULO`

---

1: **Input:** decision set $\mathcal{K}$, horizon $T$, block size $K$, step size $\eta$, error tolerance $\epsilon$, number of agents $N$, weight matrix $A = [a_{ij}]$, transformation map $h(\cdot)$, linearizable query oracle $\mathcal{G}$
2: Set $\mathbf{x}_1^i = \tilde{\mathbf{y}}_1^i = \mathbf{c} \in \mathcal{K}$ for any $i = 1, \cdots, N$
3: **for** $m = 1, \cdots, T/K$ **do**
4:     **for** each agent $i = 1, \cdots, N$ in parallel **do**
5:         **for** $t \in \mathcal{T}_m = \{(m-1)K+1, \ldots, mK\}$ **do**
6:             Play $\hat{\mathbf{x}}_t^i = h(\mathbf{x}_m^i)$
7:             Query the linearizable query oracle $\mathcal{G}$ at $\mathbf{x}_m^i$ and get response $\mathbf{o}_t^i = \mathcal{G}(\mathbf{x}_m^i)$
8:         **end for**
9:         Communicate local state $\mathbf{s}_m^i = (\mathbf{x}_m^i, \tilde{\mathbf{y}}_m^i)$ with its neighbors $\mathcal{N}_i$
10:        $\mathbf{y}_{m+1}^i \leftarrow \sum_{j \in \mathcal{N}_i} a_{ij} \tilde{\mathbf{y}}_m^j + \eta \sum_{t \in \mathcal{T}_m} \mathbf{o}_t^i$
11:        $(\mathbf{x}_{m+1}^i, \tilde{\mathbf{y}}_{m+1}^i) \leftarrow \mathcal{O}_{IP}(\mathcal{K}, \sum_{j \in \mathcal{N}_i} a_{ij} \mathbf{x}_m^j, \mathbf{y}_{m+1}^i, \epsilon)$
12:     **end for**
13: **end for**

---

## 4.2 Result and Analysis

We will provide the key results for the proposed algorithm, including the regret, communication complexity, and the number of total LOO calls used by the proposed algorithm. The result is given in Theorem 1.

**Theorem 1.** *Given Assumptions 1, 2, 3, 4, Algorithm 1 ensures that the $\alpha$-regret for agent $i$ is bounded as*

$$\mathbb{E}\left[\mathcal{R}_\alpha^i\right] \leq \beta \left[\frac{2R^2}{\eta} + \frac{18\epsilon T}{\eta K} + 7\eta TKG^2 + 13TG\sqrt{3\epsilon}\right]$$

$$+ \frac{\beta}{1-\lambda_2}\left(\frac{12\epsilon T}{\eta K} + 9\eta TKG^2 + 12TG\sqrt{3\epsilon}\right)$$

$$+ (G + 2L_1R)(N^{1/2} + 1)\beta\left(3\sqrt{2\epsilon} + \frac{(3\eta KG + 2\sqrt{3\epsilon})}{1-\lambda_2}\right).$$

*Further, the communication complexity is $O(T/K)$. Finally, the number of LOO calls are upper bounded as $\frac{27TR^2}{\epsilon K}\left(8.5 + 5.5\frac{K^2\eta^2 G^2}{\epsilon} + \frac{K^4\eta^4 G^4}{\epsilon^2}\right)$. In particular, if we set $\epsilon = K^2\eta^2 G^2$, then we have*

$$\mathbb{E}\left[\mathcal{R}_\alpha^i\right] = O\left(\frac{1}{\eta} + \eta TKG^2\right),$$

*and the number of LOO calls is $O(\frac{T}{\epsilon K})$.*

*Proof.* The detailed proof of Theorem 1 is provided in the Appendix C. For the completeness of argument, we provide a high-level outline of proof for the regret bound and the communication complexity.

**Regret:** Note that instead of a 1-weakly DR-submodular function which has the nice property of $\nabla f(\mathbf{x}) \geq f(\mathbf{y}), \forall \mathbf{x} \leq \mathbf{y}$, we are dealing with upper linearizable functions, a much more generalized function class that includes any function that satisfies $\alpha f(\mathbf{y}) - f(h(\mathbf{x})) \leq \beta\left(\langle \mathfrak{g}(f,\mathbf{x}), \mathbf{y} - \mathbf{x}\rangle\right)$ for some functional $\mathfrak{g}$, function $h$, and constants $0 < \alpha \leq 1$ and $\beta > 0$ as previously described, of which 1-weakly DR-submodular function is an instance.

As is common to bound regret of convex algorithm with first order linear approximation (Orabona, 2019), we bound the regret using the inner product of the distance in action space and $\mathfrak{g}$ function space to approximate the function value, with the help of law of iterated expectation. Let $\mathbf{x}^* \in \text{argmax}_{\mathbf{u}\in\mathcal{K}} \frac{1}{N}\sum_{t=1}^T \sum_{i=1}^N f_{t,i}(\mathbf{u})$ denote the optimal action, we have

$$\mathbb{E}[\mathcal{R}_\alpha^j] = \frac{1}{N}\sum_{i=1}^N \sum_{m=1}^{T/K} \sum_{t\in\mathcal{T}_m} \mathbb{E}\left[\alpha f_{t,i}(\mathbf{x}^*) - f_{t,i}(h(\mathbf{x}_m^j))\right]$$

$$\leq \frac{\beta}{N}\sum_{i=1}^N \sum_{m=1}^{T/K} \sum_{t\in\mathcal{T}_m} \mathbb{E}\left[\langle \mathbf{x}^* - \mathbf{x}_m^j, \tilde{\mathfrak{g}}_{t,i}(\mathbf{x}_m^j)\rangle\right]$$

Rearranging terms to better leverage the communication structure of the decentralized network, we have:

$$\frac{1}{\beta}\mathbb{E}[\mathcal{R}_\alpha^j] \leq \mathbb{E}\left[\frac{1}{N}\sum_{i=1}^N \sum_{m=1}^{T/K} \sum_{t\in\mathcal{T}_m} \langle \mathbf{x}^* - \mathbf{x}_m^i, \tilde{\mathfrak{g}}_{t,i}(\mathbf{x}_m^i)\rangle\right]$$

$$+ \mathbb{E}\left[\frac{1}{N}\sum_{i=1}^N \sum_{m=1}^{T/K} \sum_{t\in\mathcal{T}_m} \langle \mathbf{x}_m^i - \mathbf{x}_m^j, \tilde{\mathfrak{g}}_{t,i}(\mathbf{x}_m^i)\rangle\right] \tag{4}$$

$$+ \mathbb{E}\left[\frac{1}{N}\sum_{i=1}^N \sum_{m=1}^{T/K} \sum_{t\in\mathcal{T}_m} \langle \mathbf{x}^* - \mathbf{x}_m^j, \tilde{\mathfrak{g}}_{t,i}(\mathbf{x}_m^j) - \tilde{\mathfrak{g}}_{t,i}(\mathbf{x}_m^i)\rangle\right]$$

Let the three parts in Equation (4) be respectively $P_1$, $P_2$, $P_3$. Through exploitation of properties of the loss functions and domain, the update rule (line 10) and the infeasible projection operation (line 11) in Algorithm 1, we obtain the upper bound of the expectation of each part:

$$\mathbb{E}\left[P_1\right] \leq \frac{R^2}{\eta} + \frac{18\epsilon T}{\eta K} + 7\eta T K G^2 + 13 T G \sqrt{3\epsilon}$$
$$+ \frac{1}{1-\lambda_2}\left(\frac{12\epsilon T}{\eta K} + 9\eta T K G^2 + 12 T G \sqrt{3\epsilon}\right)$$
$$\mathbb{E}\left[P_2\right] \leq G(N^{1/2} + 1)\left(3\sqrt{2\epsilon} + \frac{3\eta K G + 2\sqrt{3\epsilon}}{1-\lambda_2}\right)$$
$$\mathbb{E}\left[P_3\right] \leq 2L_1 R(N^{1/2} + 1)\left(3\sqrt{2\epsilon} + \frac{3\eta K G + 2\sqrt{3\epsilon}}{1-\lambda_2}\right)$$

Adding $P_1, P_2, P_3$, we obtain the upper bound for $\alpha$-regret for agent $i$ as given in the statement of the Theorem.

**LOO calls:** Based on Lemma 1 for the infeasible projection oracle, we have the number of LOO calls for agent $i$ in block $m$ as:

$$l_m^i = \frac{27R^2}{\epsilon}\max\left(\frac{1}{4\epsilon^2}(\|\mathbf{y}_{m+1}^i - \sum_{j\in\mathcal{N}_i} a_{ij}\mathbf{x}_m^j\|^2)(\|\mathbf{y}_{m+1}^i - \sum_{j\in\mathcal{N}_i} a_{ij}\mathbf{x}_m^j\|^2 - \epsilon) + 1, 1\right) \tag{5}$$

Through exploitation of the update rule (line 10) and the infeasible projection operation (line 11) in Algorithm 1, we have

$$\left\|\mathbf{y}_{m+1}^i - \sum_{j\in\mathcal{N}_i} a_{ij}\mathbf{x}_m^j\right\|^2 \leq 2\eta^2 K^2 G^2 + 6\epsilon \tag{6}$$

Substituting (6) to (5), we obtain the total LOO calls, $\sum_{m=1}^{T/K} l_m^i$, as in the statement of the Theorem. $\square$

With appropriate selection of parameters block size $K$, update step $\eta$, and infeasible projection error tolerance $\epsilon$, we have final results for the main Algorithm 1 in Theorem 2. Motivated by the trade-off between block size and the time complexity, we introduce a hyper parameter $\theta$, through which users adjust block size accordingly, allowing resilience against practical communication limitations.

**Theorem 2.** *For Theorem 1, choosing $K = T^{1-\theta}$, $\eta = \frac{1}{\sqrt{KT}}$, and $\epsilon = K^2\eta^2$, we get that for each agent $i$ the*

$$\mathbb{E}\left[R_\alpha^i\right] = O(T^{1-\theta/2}) \tag{7}$$

*Further, the communication complexity is $O(T^\theta)$ and the number of LOO calls is $O(T^{2\theta})$.*

We note that in the special case of $\theta = 1$, there will be no block effect, and we achieve a regret of $O(\sqrt{T})$, with a communication complexity of $O(T)$ and number of LOO calls of $O(T^2)$. Further, in the special case of $\theta = 1/2$, we achieve a regret of $O(T^{3/4})$, with a communication complexity of $O(\sqrt{T})$ and number of LOO calls of $O(T)$.

## 5 Extension to Different Feedback for Up-Concave (DR-Submodular) Optimization

In Section 4, we presented DROCULO, a general algorithm for decentralized online optimization of upper-linearizable functions. The analysis assumed access to a linearizable query oracle G built upon a first-order oracle. However, this assumption may not hold in settings with more limited feedback, such as semi-bandit or bandit scenarios. This section extends our framework to address these settings for three prominent classes of up-concave functions, which are instances of the upper-linearizable class (detailed in Appendix B). We present

a series of new algorithms that adapt `DROCULO` to handle diverse feedback types. These algorithms provide the first known results for these function classes in their respective feedback settings. Notably, while prior work such as Liao et al. (2023) has addressed monotone up-concave functions over convex sets containing the origin, it was restricted to the 1-weakly up-concave case. Our results apply to the more general $\gamma$-weakly up-concave, and consider other settings including monotone and non-monotone, general convex set and convex set containing the origin.

The structure of this section results from the nature of the query required by the base algorithm. Section 5.1 addresses monotone up-concave optimization over general convex sets (Case B.1), where `DROCULO` requires only trivial queries (semi-bandit feedback). Section 5.2 addresses the other two cases (B.2 and B.3), where `DROCULO` requires non-trivial queries (full-information feedback). To achieve these extensions, we adapt the meta-algorithms from Pedramfar & Aggarwal (2024a) to the decentralized context. Similar to Section 4, we introduce assumptions about up-concave functions we consider for our extension algorithms. Assumption 3 will reduce to Assumption 3*, and Assumption 4 will reduce to Assumption 4*.

**Assumption** 3*. *We assume all $\gamma$-weakly up-concave (as defined in Equation 1) objective functions are continuous, differentiable, Lipschitz continuous, and smooth.*

**Assumption** 4*. *Given the objective functions, we assume the query oracles, whether zeroth-order or first-order, are bounded.*

## 5.1 Monotone up-concave optimization over general convex set (B.1)

For monotone up-concave functions over a general convex set(Case B.1), the transformation map $h(\cdot)$ is the identity function. Consequently, the linearizable query oracle $\mathcal{G}$ queries the first-order oracle at the point of action ($\mathbf{w}_t^i = \hat{\mathbf{x}}_t^i$), which constitutes a trivial query. This implies that for Case B.1, Algorithm 1 operates under semi-bandit feedback. To handle the more restrictive bandit feedback (when zeroth-order instead of first-order oracle is provided), we adapt the Semi-bandit To Bandit (`STB`) meta-algorithm from Pedramfar & Aggarwal (2024b) to develop Algorithm 2.

Before detailing steps in Algorithm 2, we introduce several mathematical notations that are being used by the `STB` meta-algorithm. Recall we have defined affine hull and relative interior of set $\mathcal{K}$ in Section 3. We choose a point $\mathbf{c} \in \text{relint}(\mathcal{K})$ and a real number $r > 0$ such that $\text{aff}(\mathcal{K}) \cap \mathbb{B}_r(\mathbf{c}) \subseteq \mathcal{K}$. Then, for any shrinking parameter $0 \le \delta < r$, we define $\hat{\mathcal{K}}_\delta := (1 - \frac{\delta}{r})\mathcal{K} + \frac{\delta}{r}\mathbf{c}$. For a function $f : \mathcal{K} \to \mathbb{R}$ defined on a convex set $\mathcal{K} \subseteq \mathbb{R}^d$, its $\delta$-smoothed version $\hat{f}_\delta : \hat{\mathcal{K}}_\delta \to \mathbb{R}$ is given as

$$\hat{f}_\delta(\mathbf{x}) := \mathbb{E}_{\mathbf{z} \sim \text{aff}(\mathcal{K}) \cap \mathbb{B}_\delta(\mathbf{x})}[f(\mathbf{z})] = \mathbb{E}_{\mathbf{v} \sim \mathcal{L}_0 \cap \mathbb{B}_1(\mathbf{0})}[f(\mathbf{x} + \delta\mathbf{v})],$$

where $\mathcal{L}_0 = \text{aff}(\mathcal{K}) - \mathbf{x}$, for any $\mathbf{x} \in \mathcal{K}$, is the linear space that is a translation of the affine hull of $\mathcal{K}$ and $\mathbf{v}$ is sampled uniformly at random from the $k = \dim(\mathcal{L}_0)$-dimensional ball $\mathcal{L}_0 \cap \mathbb{B}_1(\mathbf{0})$. Thus, the function value $\hat{f}_\delta(\mathbf{x})$ is obtained by "averaging" $f$ over a sliced ball of radius $\delta$ around $\mathbf{x}$. For a function class $\mathbf{F}$ over $\mathcal{K}$, we use $\hat{\mathbf{F}}_\delta$ to denote $\{\hat{f}_\delta \mid f \in \mathbf{F}\}$. We will drop the subscript $\delta$ when there is no ambiguity.

The important property of this notion is that it allows for construction of a one-point gradient estimator.

Specifically, it is known[5] that

$$\nabla \hat{f}_\delta(\mathbf{x}) = \frac{k}{\delta}\mathbb{E}_{\mathbf{v} \sim \mathcal{L}_0 \cap \mathbb{S}^1}[f(\mathbf{x} + \delta\mathbf{v})].$$

This allows us to convert Algorithm 1 to allow for zeroth order feedback. Specifically, we run Algorithm 1 on functions $\hat{f}_{t,i}$ instead of $f_{t,i}$ and when it requires an unbiased estimate of the gradient of $\hat{f}_{t,i}(\mathbf{x})$, we use $f_{t,i}(\mathbf{x} + \delta\mathbf{v})$ where $\mathbf{v}$ is sampled uniformly from $\mathcal{L}_0 \cap \mathbb{S}^1$. More generally, if we have access to $o_{t,i}$, an unbiased estimate of $f_{t,i}(\mathbf{x} + \delta\mathbf{v})$, then $\frac{k}{\delta}o_{t,i}$ is an unbiased estimate of $\nabla \hat{f}_{t,i}(\mathbf{x})$. If the zeroth order oracle, from which $\mathbf{o}_{t,i}$ is sampled, is bounded by $B_0$, then we see that this new one-point gradient estimator of $\nabla \hat{f}_{t,i}(\mathbf{x})$ is

---

[5]When $k = d$ and therefore $\mathcal{L}_0 = \mathbb{R}^d$, this equality is well known, e.g. see Nemirovskiĭ & Ĭŭdin (1983); Flaxman et al. (2005). The more general case where $k \le d$ may be found in Remark 4 in Pedramfar et al. (2023).

bounded by $G' := \frac{k}{\delta}B_0$. Therefore, if we set $\epsilon = K^2\eta^2(G')^2$, we may use Theorem 1 to see that the regret is bounded by $O\left(\frac{1}{\eta} + \eta TK(G')^2\right)$, and the number of LOO calls is $O(\frac{T}{\epsilon K})$. However, it should be noted that the functions $\hat{f}_{t,i}$ are defined over $\mathcal{K}_\delta$ and this regret is computed against the best point in $\mathcal{K}_\delta$ which can be $O(\delta)$ away from the best point in $\mathcal{K}$. Hence, we see that

$$\mathbb{E}\left[R_\alpha^i\right] = O\left(\frac{1}{\eta} + \eta TK(G')^2 + \delta T\right) = O\left(\frac{1}{\eta} + \eta TK\delta^{-2} + \delta T\right).$$

Putting these results together, we obtain the following result, with detailed discussion and proof in Appendix D.

---

**Algorithm 2** Bandit Algorithm for Case B.1

---

1: **Input:** decision set $\mathcal{K}$, horizon $T$, block size $K$, step size $\eta$, error tolerance $\epsilon$, number of agents $N$, weight matrix $\mathbf{A} = [a_{ij}]$, transformation map $h(\cdot)$, smoothing parameter $\delta \le \alpha$, shrunk set $\hat{\mathcal{K}}_\delta$, linear space $\mathcal{L}_0$, zeroth order oracle $\mathcal{Q}$
2: Let $k = \dim(\mathcal{L}_0)$
3: Set $\mathbf{x}_1^i = \tilde{\mathbf{y}}_1^i = \mathbf{c} \in \hat{\mathcal{K}}_\delta$ for any $i = 1, \cdots, N$
4: **for** $m = 1, \cdots, T/K$ **do**
5:     **for** each node $i = 1, \cdots, N$ in parallel **do**
6:         **for** $t = (m-1)K + 1, \ldots, mK$ **do**
7:             Sample $\mathbf{v}_t^i \in \mathbb{S}^1 \cap \mathcal{L}_0$ uniformly
8:             Play $\hat{\mathbf{x}}_t^i = h(\mathbf{x}_m^i) + \delta\mathbf{v}_t^i$
9:             Query the oracle $\mathcal{Q}$ at $\hat{\mathbf{x}}_t^i$ and get response $\mathbf{o}_t^i$
10:            Let $\mathbf{o}_t^i \leftarrow \frac{k}{\delta}\mathbf{o}_t^i\mathbf{v}_t^i$
11:         **end for**
12:         Communicate $\mathbf{x}_m^i$ and $\tilde{\mathbf{y}}_m^i$ with neighbors
13:         $\mathbf{y}_{m+1}^i \leftarrow \sum_{j \in \mathcal{N}_i} a_{ij}\tilde{\mathbf{y}}_m^j + \eta \sum_{t \in \mathcal{T}_m} \mathbf{o}_t^i$
14:         $(\mathbf{x}_{m+1}^i, \tilde{\mathbf{y}}_{m+1}^i) \leftarrow O_{IP}\left(\hat{\mathcal{K}}_\delta, \sum_{j \in \mathcal{N}_i} a_{ij}\mathbf{x}_m^j, \mathbf{y}_{m+1}^i, \epsilon\right)$
15:     **end for**
16: **end for**

---

**Theorem 3.** *For Case B.1, under Assumption 1, 2, 3\*, 4\*, if we set $\epsilon = K^2\eta^2\delta^{-2}$, then Algorithm 2 ensures a regret bound of*

$$\mathbb{E}\left[R_\alpha^i\right] = O\left(\frac{1}{\eta} + \eta TK\delta^{-2} + \delta T\right),$$

*with at most $O(\frac{T}{\epsilon K})$ LOO calls and $O(T/K)$ communication complexity. In particular, if we set $K = T^{1-\theta}$, $\delta = T^{-\theta/4}$ and $\eta = \frac{\delta}{\sqrt{KT}}$, we see that $\mathbb{E}\left[R_\alpha^i\right] = O(T^{1-\theta/4})$ with at most $O(T^{2\theta})$ LOO calls and $O(T^\theta)$ communication complexity.*

*Remark* 3. Our approach to gradient estimation from zeroth-order feedback relies on the one-point gradient estimator (Flaxman et al., 2005). An alternative, common in the bandit optimization literature (Agarwal et al., 2010; Shamir, 2017), is to use a two-point estimator, which queries the function at two points (e.g., $f(\mathbf{x}+\delta\mathbf{v})$ and $f(\mathbf{x}-\delta\mathbf{v})$) to construct a finite-difference approximation of the true gradient. While a two-point estimator can provide a better estimate of the gradient, it requires access to an exact value oracle, which is often an impractical assumption. If an exact value oracle is available, one could use an approach similar to Pedramfar & Aggarwal (2024a, Algorithm 7) to develop a counterpart of Algorithm 2 using the two-point gradient estimator. Note that such an algorithm will not be bandit, as it requires two queries per timestep. However, we may use arguments similar to Pedramfar & Aggarwal (2024a, Corollary 5) to see that such an algorithm has the same order of regret as the first-order algorithm it is based on. In other words, we obtain a regret bound with the same order as Theorem 2.

## 5.2 Monotone up-concave optimization over convex set containing the origin (Appendix B.2) and Non-monotone up-concave optimization over general convex set (Appendix B.3)

We now consider monotone up-concave optimization over convex sets containing the origin (Case B.2) and non-monotone up-concave optimization over general convex sets (Case B.3). In both cases, the linearizable query oracle $\mathcal{G}$ must query the first-order query oracle at a point $\mathbf{w}_t^i$ that is different from the action point $\hat{\mathbf{x}}_t^i = h(\mathbf{x}_m^i)$. This constitutes a non-trivial query, meaning that Algorithm 1 requires first-order full-information feedback for these function classes. The remainder of this section will introduce algorithms that extend our framework to handle other feedback settings for these two cases, including semi-bandit, zeroth-order full-information, and bandit feedback.

### 5.2.1 Semi-bandit Feedback

As established, Algorithm 1 requires first order full-information feedback for Case B.2 and Case B.3. To handle semi-bandit feedback (when first-order query oracle only allow trivial queries), we use the "Stochastic Full-information To Trivial query" (SFTT) meta-algorithm from Pedramfar & Aggarwal (2024a). The key challenge lies in composing the SFTT blocking mechanism with the existing block structure of DROCULO and its inter-node communication protocol. The resulting method is presented in Algorithm 3.

---

**Algorithm 3** Semi-Bandit Algorithm for Case B.2 and B.3

---

1: **Input:** decision set $\mathcal{K}$, horizon $T$, DROCULO block size $K$, step size $\eta$, error tolerance $\epsilon$, number of agents $N$, weight matrix $\mathbf{A} = [a_{ij}]$, map $h(\cdot)$, SFTT block size $L > 1$, first-order oracle $\mathcal{Q}$
2: Set $\mathbf{x}_1^i = \tilde{\mathbf{y}}_1^i = \mathbf{c} \in \mathcal{K}$ for any $i = 1, \cdots, N$
3: **for** $m = 1, \cdots, \frac{T}{LK}$ **do**
4:   **for** each node $i = 1, \cdots, N$ in parallel **do**
5:     **for** q = $(m-1)K+1, \cdots, mK$ **do**
6:       Play $\hat{\mathbf{x}}_q^i = h(\mathbf{x}_m^i)$
7:       For B.2, we sample $z$ as described in Lemma 5 and for B.3, we sample $z$ as described in Lemma 7
8:       For case B.2, let $\mathbf{w}_q^i = z * \mathbf{x}_m^i$ and for case B.3, let $\mathbf{w}_q^i = \frac{z}{2} * (\mathbf{x}_m^i - \underline{\mathbf{u}}) + \underline{\mathbf{u}}$ {$\underline{\mathbf{u}}$ is a given constant}
9:       Sample $t_q'$ uniformly from $\{(q-1)L+1, \ldots, qL\}$
10:       **for** $t = (q-1)L+1, \ldots, qL$ **do**
11:         **if** $t = t_q'$ **then**
12:           Play the action $\mathbf{z}_t^i = \mathbf{w}_q^i$
13:           Query the oracle $\mathcal{Q}$ at $\mathbf{w}_q^i$ and get response $\mathbf{o}_q^i$
14:         **else**
15:           Play the action $\mathbf{z}_t^i = \hat{\mathbf{x}}_q^i$
16:         **end if**
17:       **end for**
18:     **end for**
19:     Communicate $\mathbf{x}_m^i$ and $\tilde{\mathbf{y}}_m^i$ with neighbors
20:     $\mathbf{y}_{m+1}^i \leftarrow \sum_{j \in \mathcal{N}_i} a_{ij} \tilde{\mathbf{y}}_m^j + \eta \sum_{q=(m-1)K+1}^{mK} \mathbf{o}_q^i$
21:     $(\mathbf{x}_{m+1}^i, \tilde{\mathbf{y}}_{m+1}^i) \leftarrow \mathcal{O}_{IP}(\mathcal{K}, \sum_{j \in \mathcal{N}_i} a_{ij} \mathbf{x}_m^j, \mathbf{y}_{m+1}^i, \epsilon)$
22:   **end for**
23: **end for**

---

Let $L \geq 1$ be an integer. The main idea here is to consider the functions $(\bar{f}_{q,i})_{1 \leq q \leq T/L, 1 \leq i \leq N}$ where $\bar{f}_{q,i} = \frac{1}{L} \sum_{t=(q-1)L+1}^{qL} f_{t,i}$. We want to run Algorithm 1 against this sequence of functions. To do this, we need to construct unbiased estimates of the gradient of $\bar{f}_{q,i}$. This can be achieved by considering a random permutation $t_1', \cdots, t_L'$ of $(q-1)L+1, \cdots, qL$ and picking $f_{t_1',i}$. Since we want an algorithm with semi-bandit feedback, at time-step $t_1'$ we select the point where the original algorithm, i.e., Algorithm 1, needed to query. In the other $L-1$ time-steps within this block, we pick the action that Algorithm 1 wants to take and ignore the returned value of the query function. Thus, at one time-step per each block of length $L$, we

have no control over the regret, which adds $O(T/L)$ to the total regret. In the remaining time-steps, the behavior is similar to Algorithm 1, with each action repeated $L-1$ times. We note that we are running Algorithm 1 against $\bar{f}_{q,i}$ with a horizon of $T' := T/L$. Hence, using the discussion above and Theorem 1, if we set $\epsilon = K^2\eta^2G^2$, then we see that the regret is bounded by $\mathbb{E}\left[R_\alpha^i\right] = (L-1)O\left(\frac{1}{\eta} + \eta T'KG^2\right) + O\left(\frac{T}{L}\right)$, the number of LOO calls is $O(\frac{T'}{\epsilon K})$, and the communication complexity is bounded by $O(T'/K)$. The key result is summarized in Theorem 4, and the detailed discussion and proof can be found in Appendix E.

**Theorem 4.** *For Case B.2 and B.3, under Assumption 1, 2, 3\*, 4\*, if we set $\epsilon = K^2\eta^2G^2$, then Algorithm 3 ensures a regret bound of*

$$\mathbb{E}\left[R_\alpha^i\right] = O\left(\frac{L}{\eta} + \eta TKG^2 + \frac{T}{L}\right).$$

*with at most $O(\frac{T}{\epsilon KL})$ LOO calls and $O(\frac{T}{KL})$ communication complexity. In particular, if $0 \le \theta \le 2/3$ and we set $K = T^{1-3\theta/2}$, $L = T^{\theta/2}$, and $\eta = T^{\theta-1}$, we see that $\mathbb{E}\left[R_\alpha^i\right] = O(T^{1-\theta/2})$ with at most $O(T^{2\theta})$ LOO calls and $O(T^\theta)$ communication complexity.*

### 5.2.2 Zeroth-order Full-information Feedback

To adapt `DROCULO` for zeroth-order full-information feedback, where only function values can be queried, we employ a strategy analogous to the one in Section 5.1. Specifically, we adapt the "First Order To Zeroth Order" (`FOTZO`) meta-algorithm from Pedramfar & Aggarwal (2024a).

The core of this method is function smoothing. Instead of operating on the original functions $f_{t,i}$, the algorithm operates on their smoothed versions, $\hat{f}_{t,i}$. This allows us to construct a one-point gradient estimator. By querying the function value at a randomly perturbed point $\mathbf{w}_t^i + \delta\mathbf{v}_t^i$, we can obtain an unbiased estimate of the gradient $\nabla \hat{f}_{t,i}(\mathbf{w}_t^i)$. This estimated gradient can then be used in place of the true gradient required by the original `DROCULO` algorithm. The resulting procedure is detailed in Algorithm 4. The introduction of the smoothing parameter $\delta$ affects the regret analysis, adding terms dependent on $\delta$ to account for the approximation error. A full analysis and proof are provided in Appendix F, leading to the regret bound stated in Theorem 5.

---

**Algorithm 4** Zeroth-order Full-information Algorithm for Case B.2 and B.3

---

1: **Input:** decision set $\mathcal{K}$, horizon $T$, `DROCULO` block size $K$, step size $\eta$, error tolerance $\epsilon$, number of agents $N$, weight matrix $\mathbf{A} = [a_{ij}]$, map $h(\cdot)$, smoothing parameter $\delta \le \alpha$, shrunk set $\hat{\mathcal{K}}_\delta$, linear space $\mathcal{L}_0$, zeroth-order query oracle $\mathcal{Q}$
2: Let $k = \dim(\mathcal{L}_0)$
3: Set $\mathbf{x}_1^i = \tilde{\mathbf{y}}_1^i = \mathbf{c} \in \hat{\mathcal{K}}_\delta$ for any $i = 1, \cdots, N$
4: **for** $m = 1, \cdots, T/K$ **do**
5:   **for** each node $i = 1, \cdots, N$ in parallel **do**
6:     **for** $t = (m-1)K+1, \ldots, mK$ **do**
7:       Play $h(\mathbf{x}_m^i)$
8:       Sample $\mathbf{v}_t^i \in \mathbb{S}^1 \cap \mathcal{L}_0$ uniformly
9:       For case B.2, let $\mathbf{w}_q^i = z * \mathbf{x}_m^i$ and for case B.3, let $\mathbf{w}_q^i = \frac{z}{2} * (\mathbf{x}_m^i - \underline{\mathbf{u}}) + \underline{\mathbf{u}}$
10:       Query the oracle $\mathcal{Q}$ at $\mathbf{w}_t^i + \delta\mathbf{v}_t^i$ and get response $\mathbf{o}_t^i$
11:       Let $\mathbf{o}_t^i \leftarrow \frac{k}{\delta}\mathbf{o}_t^i\mathbf{v}_t^i$
12:     **end for**
13:     Communicate $\mathbf{x}_m^i$ and $\tilde{\mathbf{y}}_m^i$ with neighbors
14:     $\mathbf{y}_{m+1}^i \leftarrow \sum\limits_{j \in \mathcal{N}_i} a_{ij}\tilde{\mathbf{y}}_m^j + \eta \sum\limits_{t \in \mathcal{T}_m} \mathbf{o}_t^i$
15:     $(\mathbf{x}_{m+1}^i, \tilde{\mathbf{y}}_{m+1}^i) \leftarrow \mathcal{O}_{IP}(\hat{\mathcal{K}}_\delta, \sum\limits_{j \in \mathcal{N}_i} a_{ij}\mathbf{x}_m^j, \mathbf{y}_{m+1}^i, \epsilon)$
16:   **end for**
17: **end for**

---

**Theorem 5.** *For Case B.2 and B.3, under Assumption 1, 2, 3\*, 4\*, if we set $\epsilon = K^2\eta^2\delta^{-2}$, then Algorithm 4 ensures a regret bound of*

$$\mathbb{E}\left[R^i_\alpha\right] = O\left(\frac{1}{\eta} + \eta T K \delta^{-2} + \delta T\right),$$

*with at most $O(\frac{T}{\epsilon K})$ LOO calls and $O(T/K)$ communication complexity. In particular, if we set $K = T^{1-\theta}$, $\delta = T^{-\theta/4}$ and $\eta = \frac{\delta}{\sqrt{KT}}$, we see that $\mathbb{E}\left[R^i_\alpha\right] = O(T^{1-\theta/4})$ with at most $O(T^{2\theta})$ LOO calls and $O(T^\theta)$ communication complexity.*

Similar to Remark 3, if an exact value oracle for the objective function is available, one can develop an counterpart of Algorithm 4 using two-point query instead of one-point query, and obtain a regret at the same order of Theorem 2.

---

**Algorithm 5** Bandit Algorithm for Case B.2 and B.3

---

1: **Input:** decision set $\mathcal{K}$, horizon $T$, `DROCULO` block size $K$, step size $\eta$, error tolerance $\epsilon$, number of agents $N$, weight matrix $\mathbf{A} = [a_{ij}]$, map $h(\cdot)$, `SFTT` block size $L > 1$, smoothing parameter $\delta \leq \alpha$, shrunk set $\hat{\mathcal{K}}_\delta$, linear space $\mathcal{L}_0$, zeroth-order query oracle $\mathcal{Q}$
2: Let $k = \dim(\mathcal{L}_0)$
3: Set $\mathbf{x}^i_1 = \tilde{\mathbf{y}}^i_1 = \mathbf{c} \in \hat{\mathcal{K}}_\delta$ for any $i = 1, \cdots, N$
4: **for** $m = 1, \cdots, T/LK$ **do**
5:     **for** each node $i = 1, \cdots, N$ in parallel **do**
6:         **for** q = $1, 2, \ldots, K$ **do**
7:             Let $\hat{\mathbf{x}}^i_q = h(\mathbf{x}^i_m)$
8:             Sample $\mathbf{v}^i_q \in \mathbb{S}^1 \cap \mathcal{L}_0$ uniformly
9:             For case B.2, let $\mathbf{w}^i_q = z * \mathbf{x}^i_m$ and for case B.3, let $\mathbf{w}^i_q = \frac{z}{2} * (\mathbf{x}^i_m - \underline{\mathbf{u}}) + \underline{\mathbf{u}}$
10:             Let $\hat{\mathbf{w}}^i_q = \mathbf{w}^i_q + \delta\mathbf{v}^i_q$
11:             Sample $t'_q$ uniformly from $\{(m-1)KL + (q-1)L + 1, \ldots, (m-1)KL + qL\}$
12:             **for** $t = (m-1)KL + (q-1)L + 1, \ldots, (m-1)KL + qL$ **do**
13:                 **if** $t = t'_q$ **then**
14:                     Play the action $\mathbf{z}_t = \hat{\mathbf{w}}^i_q$
15:                     Query the oracle $\mathcal{Q}$ at $\hat{\mathbf{w}}^i_q$ and get response $\mathbf{o}^i_q$
16:                     Let $\mathbf{o}^i_q \leftarrow \frac{k}{\delta}\mathbf{o}^i_q\mathbf{v}^i_q$
17:                 **else**
18:                     Play the action $\mathbf{z}_t = \hat{\mathbf{x}}^i_q$
19:                 **end if**
20:             **end for**
21:         **end for**
22:         Communicate $\mathbf{x}^i_m$ and $\tilde{\mathbf{y}}^i_m$ with neighbours
23:         $\mathbf{y}^i_{m+1} \leftarrow \sum\limits_{j \in \mathcal{N}_i} a_{ij}\tilde{\mathbf{y}}^j_m + \eta \sum\limits_{q=1}^K \mathbf{o}^i_q$
24:         $(\mathbf{x}^i_{m+1}, \tilde{\mathbf{y}}^i_{m+1}) \leftarrow \mathcal{O}_{IP}(\hat{\mathcal{K}}_\delta, \sum\limits_{j \in \mathcal{N}_i} a_{ij}\mathbf{x}^j_m, \mathbf{y}^i_{m+1}, \epsilon)$
25:     **end for**
26: **end for**

---

### 5.2.3 Bandit Feedback

Finally, to design an algorithm that can operate under bandit feedback (i.e., zeroth-order, trivial queries) for case B.2 and B.3, we combine the strategies from the preceding two subsections. The process involves a two-step adaptation of the original `DROCULO` algorithm. First, we apply the `FOTZO` meta-algorithm to handle zeroth-order feedback, resulting in Algorithm 4. Second, we apply the `SFTT` meta-algorithm to Algorithm 4 to convert its non-trivial queries into trivial ones. The resulting method, presented in Algorithm 5, effectively

applies the `SFTT` blocking mechanism to the smoothed-function approach of Algorithm 4. The analysis of this composite algorithm must therefore account for error terms from both adaptations: the function smoothing (parameterized by $\delta$) and the `SFTT` blocking (parameterized by $L$). This leads to the final regret bound presented in Theorem 6, with a detailed proof available in Appendix G.

**Theorem 6.** *For Case B.2 and B.3, under Assumption 1, 2, 3\*, 4\*, if we set $\epsilon = K^2\eta^2\delta^{-2}$, then Algorithm 5 ensures a regret bound of*

$$\mathbb{E}\left[R_\alpha^i\right] = O\left(\frac{L}{\eta} + \eta T K \delta^{-2} + \delta T + \frac{T}{L}\right),$$

*with at most $O(\frac{T}{\epsilon K L})$ LOO calls and $O(\frac{T}{KL})$ communication complexity. In particular, if $0 \leq \theta \leq 4/5$ and we set $K = T^{1-5\theta/4}$, $\delta = T^{-\theta/4}$, $L = T^{\theta/4}$, and $\eta = T^{\theta/2-1}$, we see that $\mathbb{E}\left[R_\alpha^i\right] = O(T^{1-\theta/4})$ with at most $O(T^{2\theta})$ LOO calls and $O(T^\theta)$ communication complexity.*

## 6 Conclusions

In this paper, we presented a decentralized, projection-free approach to optimizing upper-linearizable functions, which extends the analysis of classical DR-submodular and concave functions. By incorporating projection-free methods, our framework provides efficient regret bounds of $O(T^{1-\theta/2})$, in first order feedback case, $O(T^{1-\theta/4})$, in zeroth order feedback case, with a communication complexity of $O(T^\theta)$ and number of linear optimization oracle calls of $O(T^{2\theta})$ for suitable choices of $0 \leq \theta \leq 1$, making it scalable for large decentralized networks. This illustrates a tradeoff between the regret and the communication complexity. The versatility of our approach allows it to handle a variety of feedback models, including full information, semi-bandit, and bandit settings. This is the first known result that provides such generalized guarantees for monotone and non-monotone up-concave functions over general convex sets. Finally, an important next step is the empirical validation of our theoretical guarantees to explore the practical performance of our framework in real-world scenarios.

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

## A    From concavity to upper-linearizability

In this section, we provide an alternative definition of upper-linearizability to further clarify the connection between this notion and that of concavity.

We start with a definition. Let $\mathcal{F}$ be a class of functions over a convex set $\mathcal{K} \subseteq \mathbb{R}^d$ and let $L$ be a functional such that, for any $f \in \mathcal{F}$ and $\mathbf{x} \in \mathcal{K}$, $L_{f,\mathbf{x}}$ is an affine map over $\mathbb{R}^d$. Let us call such a functional $L$ a linear assignment for $\mathcal{F}$.

**Lemma 2.** *A continuously differentiable function class $\mathcal{F}$ consists only of concave functions if and only if it has a linear assignment $L$ such that, for all $f \in \mathcal{F}$ and $\mathbf{x}, \mathbf{y} \in \mathcal{K}$, we have*

$$L_{f,\mathbf{x}}(\mathbf{x}) = f(\mathbf{x}) \quad and \quad L_{f,\mathbf{x}}(\mathbf{y}) \geq f(\mathbf{y}).$$

*Proof.* We provide the proof for the case where $\dim(\mathcal{K}) = d$, the more general case is similar.

If $f$ is concave, then we may choose the linear assignment $L_{f,\mathbf{x}}(\mathbf{y}) := f(\mathbf{x}) + \langle \nabla f(\mathbf{x}), \mathbf{y} - \mathbf{x} \rangle$. The fact that $L_{f,\mathbf{x}}(\mathbf{y}) \geq f(\mathbf{y})$ for all $\mathbf{y} \in \mathcal{K}$ follows from the definition of concavity.

On the other hand, if such a linear assignment exists for $f \in \mathcal{F}$, then we have

$$f(\mathbf{y}) \leq L_{f,\mathbf{x}}(\mathbf{y}) = L_{f,\mathbf{x}}(\mathbf{0}) + L_{f,\mathbf{x}}(\mathbf{y}) - L_{f,\mathbf{x}}(\mathbf{0}).$$

Since $L_{f,\mathbf{x}}$ is affine, the expression $L_{f,\mathbf{x}}(\mathbf{y}) - L_{f,\mathbf{x}}(\mathbf{0})$ is linear. Therefore, there is a vector $A \in \mathbb{R}^d$ such that $L_{f,\mathbf{x}}(\mathbf{y}) - L_{f,\mathbf{x}}(\mathbf{0}) = \langle A, \mathbf{y} \rangle$. Thus

$$f(\mathbf{y}) \leq L_{f,\mathbf{x}}(\mathbf{y}) = L_{f,\mathbf{x}}(\mathbf{x}) + (L_{f,\mathbf{x}}(\mathbf{y}) - L_{f,\mathbf{x}}(\mathbf{0})) - (L_{f,\mathbf{x}}(\mathbf{x}) - L_{f,\mathbf{x}}(\mathbf{0})) = f(\mathbf{x}) + \langle A, \mathbf{y} - \mathbf{x} \rangle.$$

Therefore, by setting $\mathbf{h} = \mathbf{y} - \mathbf{x}$ and $\mathbf{u} = \frac{\mathbf{h}}{\|\mathbf{h}\|} \in \mathbb{S}^d$, we see that

$$\frac{f(\mathbf{x} + \mathbf{h}) - f(\mathbf{x}) - \langle \nabla f(x), \mathbf{h} \rangle}{\|\mathbf{h}\|} \leq \frac{\langle A - \nabla f(x), \mathbf{h} \rangle}{\|\mathbf{h}\|} = \langle A - \nabla f(\mathbf{x}), \mathbf{u} \rangle.$$

If we keep $\mathbf{u}$ fixed and take the limit $\|\mathbf{h}\| \to 0$, the left-hand side of the above expression tends to zero, while the right-hand side remains constant. Thus, we see that

$$\langle A - \nabla f(\mathbf{x}), \mathbf{u} \rangle \geq 0.$$

If $\mathbf{x} \in \mathrm{relint}(\mathcal{K})$, then there exists a $d$-dimensional ball around $\mathbf{x}$ that is contained in $\mathcal{K}$. Thus, the above inequality holds for any choice of $\mathbf{u}$. In particular, it also holds for $-\mathbf{u}$, which implies that $\langle A - \nabla f(\mathbf{x}), \mathbf{u} \rangle = 0$, for all $\mathbf{u} \in \mathbb{S}^d$. Hence, we conclude that $A = \nabla f(x)$. Therefore, for all $\mathbf{x} \in \mathrm{relint}(\mathcal{K})$, we have

$$L_{f,\mathbf{x}}(\mathbf{y}) = L_{f,\mathbf{x}}(\mathbf{x}) + (L_{f,\mathbf{x}}(\mathbf{y}) - L_{f,\mathbf{x}}(\mathbf{0})) - (L_{f,\mathbf{x}}(\mathbf{x}) - L_{f,\mathbf{x}}(\mathbf{0})) = f(\mathbf{x}) + \langle \nabla f(\mathbf{x}), \mathbf{y} - \mathbf{x} \rangle,$$

Hence $f(\mathbf{y}) \leq f(\mathbf{x}) + \langle \nabla f(\mathbf{x}), \mathbf{y} - \mathbf{x} \rangle$ for all $\mathbf{x} \in \mathrm{relint}(\mathcal{K})$ and $\mathbf{y} \in \mathcal{K}$. Since $f$ is continuously differentiable, this inequality holds for all $\mathbf{x}, \mathbf{y} \in \mathcal{K}$ and therefore $f$ is concave.  □

Now we may phrase the definition of upper-linearizability in a way that is similar to the above lemma. We can also see clearly why such function classes are called "upper-linearizable".

**Lemma 3.** *A function class $\mathcal{F}$ is called upper-linearizable if and only if it has a linear assignment $L$ such that, for all $f \in \mathcal{F}$ and $\mathbf{x}, \mathbf{y} \in \mathcal{K}$, we have*

$$L_{f,\mathbf{x}}(\mathbf{x}) = \frac{1}{\alpha} f(h(\mathbf{x})) \quad and \quad L_{f,\mathbf{x}}(\mathbf{y}) \geq f(\mathbf{y}),$$

*for some function $h : \mathcal{K} \to \mathcal{K}$ and some $\alpha \in (0, 1]$.*

The proof is clear from the definition.

# B Up-concave Functions are Linearizable

In this section, we provide for completeness that three cases of common up-concave functions can be formulated as upper-linearizable functions, and we give the exact query algorithm to obtain estimates of $\mathfrak{g}$ functions. We further show that $\mathfrak{g}$ given by these algorithms is Lipschitz-continuous, if $f$ is $L$-smooth.

## B.1 Monotone Up-concave optimization over general convex set

Following Lemma 1 by Pedramfar & Aggarwal (2024a), we note that monotone up-concave optimization over general convex set can be formulated as an online maximization by quantization algorithm with trivial query $\mathcal{G}(\mathbf{x}) = \mathbf{x}$.

**Lemma 4** (Pedramfar & Aggarwal (2024a)). *Let $f : [0,1]^d \to \mathbb{R}$ be a non-negative monotone $\gamma$-weakly up-concave function with curvature bounded by $c$. Then, for all $\mathbf{x}, \mathbf{y} \in [0,1]^d$, we have*

$$\frac{\gamma^2}{1 + c\gamma^2} f(\mathbf{y}) - f(\mathbf{x}) \le \frac{\gamma}{1 + c\gamma^2} (\langle \nabla f(\mathbf{x}), \mathbf{y} - \mathbf{x} \rangle),$$

*where $\nabla f$ is the gradient of $f$.*

---

**Algorithm 6** BQM: Boosted Query Oracle for Monotone up-concave functions over general convex sets

---
1: **Input:** First order query oracle $\mathcal{Q}$, point $\mathbf{x}$
2: **Return:** the output of the first-order query oracle $\mathcal{Q}$ at $\mathbf{x}$

---

## B.2 Monotone up-concave optimization over convex set containing the origin

Following Lemma 2 by Pedramfar & Aggarwal (2024a), we note that monotone up-concave optimization over convex set containing the origin can be formulated as an online maximization by quantization algorithm with non-trivial query $\mathcal{G} = \texttt{BQMO}$, which is described in Algorithm 7.

**Lemma 5** (Pedramfar & Aggarwal (2024a)). *Let $f : [0,1]^d \to \mathbb{R}$ be a non-negative monotone $\gamma$-weakly up-concave differentiable function and let $F : [0,1]^d \to \mathbb{R}$ be the function defined by*

$$F(\mathbf{x}) := \int_0^1 \frac{\gamma e^{\gamma(z-1)}}{(1 - e^{-\gamma})z} (f(z * \mathbf{x}) - f(\mathbf{0})) dz.$$

*Then $F$ is differentiable and, if the random variable $\mathcal{Z} \in [0,1]$ is defined by the law*

$$\forall z \in [0,1], \quad \mathbb{P}(\mathcal{Z} \le z) = \int_0^z \frac{\gamma e^{\gamma(u-1)}}{1 - e^{-\gamma}} du, \tag{8}$$

*then we have $\mathbb{E}[\nabla f(\mathcal{Z} * \mathbf{x})] = \nabla F(\mathbf{x})$. Moreover, we have*

$$(1 - e^{-\gamma}) f(\mathbf{y}) - f(\mathbf{x}) \le \frac{1 - e^{-\gamma}}{\gamma} \langle \nabla F(\mathbf{x}), \mathbf{y} - \mathbf{x} \rangle.$$

---

**Algorithm 7** BQMO: Boosted Query Oracle for Monotone up-concave functions over convex sets containing the origin

---
1: **Input:** First order query oracle $\mathcal{Q}$, point $\mathbf{x}$
2: Sample $z \in [0,1]$ according to Equation (8) in Lemma 5
3: **Return:** the output of the first-order query oracle $\mathcal{Q}$ at $z * \mathbf{x}$

---

In this case, $\mathfrak{g} = \nabla F(\mathbf{x})$, $h(\mathbf{x}) = \text{Id}$. Further, if $f$ is smooth, $\mathfrak{g}$ is Lipschitz continuous, as shown in the following Lemma.

**Lemma 6** (Theorem 2(iii), Zhang et al. (2022)). *If $f$ is $L$-smooth and satisfies other assumptions of Lemma 5, $F$ is $L'$-smooth, where $L' = L \frac{\gamma + e^{-\gamma} - 1}{\gamma(1 - e^{-\gamma})}$, i.e., $\nabla F(\mathbf{x})$ is $L'$-Lipschitz continuous.*

### B.3 Non-monotone up-concave optimization over general convex set

Following Lemma 3 by Pedramfar & Aggarwal (2024a), we note that non-monotone up-concave optimization over general convex set can be formulated as an online maximization by quantization algorithm with non-trivial query $\mathcal{G} = \texttt{BQN}$, as described in Algorithm 8.

**Lemma 7** (Pedramfar & Aggarwal (2024a)). *Let $f : [0,1]^d \to \mathbb{R}$ be a non-negative non-monotone continuous up-concave differentiable function and let $\underline{\mathbf{x}} \in \mathcal{K}$. Define $F : [0,1]^d \to \mathbb{R}$ as the function*

$$F(\mathbf{x}) := \int_0^1 \frac{2}{3z(1 - \frac{z}{2})^3} \left( f\left( \frac{z}{2} * (\mathbf{x} - \underline{\mathbf{x}}) + \underline{\mathbf{x}} \right) - f(\underline{\mathbf{x}}) \right) dz,$$

*Then $F$ is differentiable and, if the random variable $\mathcal{Z} \in [0,1]$ is defined by the law*

$$\forall z \in [0,1], \quad \mathbb{P}(\mathcal{Z} \leq z) = \int_0^z \frac{1}{3(1 - \frac{u}{2})^3} du, \tag{9}$$

*then we have $\mathbb{E}\left[ \nabla f\left( \frac{\mathcal{Z}}{2} * (\mathbf{x} - \underline{\mathbf{x}}) + \underline{\mathbf{x}} \right) \right] = \nabla F(\mathbf{x})$. Moreover, we have*

$$\frac{1 - \|\underline{\mathbf{x}}\|_\infty}{4} f(\mathbf{y}) - f\left( \frac{\mathbf{x} + \underline{\mathbf{x}}}{2} \right) \leq \frac{3}{8} \langle \nabla F(\mathbf{x}), \mathbf{y} - \mathbf{x} \rangle.$$

---

**Algorithm 8** $\texttt{BQN}$: Boosted Query Algorithm for non-monotone up-concave functions over general convex sets

1: **Input:** First order query oracle $\mathcal{Q}$, point $\mathbf{x}$
2: Sample $z \in [0,1]$ according to Equation 9
3: **Return:** the output of first-order query oracle $\mathcal{Q}$ at $\frac{z}{2} * (\mathbf{x} - \underline{\mathbf{x}}) + \underline{\mathbf{x}}$

---

In this case, $\mathfrak{g} = \nabla F(\mathbf{x})$, $h(\mathbf{x}) : \mathbf{x} \mapsto \frac{\mathbf{x}_t + \mathbf{x}}{2}$. Further, if $f$ is $L$-smooth, $\mathfrak{g}$ is Lipschitz continuous, as given in the following Lemma.

**Lemma 8** (Theorem 18, Zhang et al. (2024)). *If $f$ is $L$-smooth, $L_1$-Lipschitz and $f$ satisfies assumptions in Lemma 7, $\nabla F(\mathbf{x})$ is $\frac{1}{8}L$-smooth and $\frac{3}{8}L_1$-Lipschitz continuous.*

## C   Proof of Theorem 1

Suppose for agent $i$ at round $t$, we denote the output of $\mathcal{G}(\mathbf{x}_m^i)$ as $\tilde{\mathfrak{g}}_{t,i}(\mathbf{x}_m^i)$. Let $\mathbf{x}^* = \operatorname{argmax}_{\mathbf{u} \in \mathcal{K}} \frac{1}{N} \sum_{t=1}^{T} \sum_{i=1}^{N} f_{t,i}(\mathbf{u})$, $\mathcal{T}_m = \{(m-1)K + 1, \cdots, mK\}$. By the definition of $\alpha$-regret for agent $j$,

we have

$$
\begin{aligned}
\mathbb{E}[\mathcal{R}_\alpha^j] &= \frac{1}{N} \sum_{t=1}^{T} \sum_{i=1}^{N} \mathbb{E}\left[\alpha f_{t,i}(\mathbf{x}^*) - f_{t,i}(h(\mathbf{x}_t^j))\right] \\
&= \frac{1}{N} \sum_{i=1}^{N} \sum_{m=1}^{T/K} \sum_{t\in\mathcal{T}_m} \mathbb{E}\left[\alpha f_{t,i}(\mathbf{x}^*) - f_{t,i}(h(\mathbf{x}_m^j))\right] \\
&\overset{(a)}{\leq} \frac{\beta}{N} \sum_{i=1}^{N} \sum_{m=1}^{T/K} \sum_{t\in\mathcal{T}_m} \mathbb{E}\left[\langle \mathbf{x}^* - \mathbf{x}_m^j, \mathfrak{g}_{t,i}(\mathbf{x}_m^j)\rangle\right] \\
&\overset{(b)}{=} \frac{\beta}{N} \sum_{i=1}^{N} \sum_{m=1}^{T/K} \sum_{t\in\mathcal{T}_m} \mathbb{E}\left[\langle \mathbf{x}^* - \mathbf{x}_m^j, \mathbb{E}\left[\tilde{\mathfrak{g}}_{t,i}(\mathbf{x}_m^j)|\mathbf{x}_m^j\right]\rangle\right] \\
&\overset{(c)}{=} \frac{\beta}{N} \sum_{i=1}^{N} \sum_{m=1}^{T/K} \sum_{t\in\mathcal{T}_m} \mathbb{E}\left[\mathbb{E}\left[\langle \mathbf{x}^* - \mathbf{x}_m^j, \tilde{\mathfrak{g}}_{t,i}(\mathbf{x}_m^j)\rangle|\mathbf{x}_m^j\right]\right] \\
&\overset{(d)}{=} \frac{\beta}{N} \sum_{i=1}^{N} \sum_{m=1}^{T/K} \sum_{t\in\mathcal{T}_m} \mathbb{E}\left[\langle \mathbf{x}^* - \mathbf{x}_m^j, \tilde{\mathfrak{g}}_{t,i}(\mathbf{x}_m^j)\rangle\right] \\
&\overset{(e)}{=} \mathbb{E}\left[\underbrace{\frac{\beta}{N} \sum_{i=1}^{N} \sum_{m=1}^{T/K} \sum_{t\in\mathcal{T}_m} \langle \mathbf{x}^* - \mathbf{x}_m^i, \tilde{\mathfrak{g}}_{t,i}(\mathbf{x}_m^i)\rangle}_{:=P_1}\right] + \mathbb{E}\left[\underbrace{\frac{\beta}{N} \sum_{i=1}^{N} \sum_{m=1}^{T/K} \sum_{t\in\mathcal{T}_m} \langle \mathbf{x}_m^i - \mathbf{x}_m^j, \tilde{\mathfrak{g}}_{t,i}(\mathbf{x}_m^i)\rangle}_{:=P_2}\right] \\
&\quad + \mathbb{E}\left[\underbrace{\frac{\beta}{N} \sum_{i=1}^{N} \sum_{m=1}^{T/K} \sum_{t\in\mathcal{T}_m} \langle \mathbf{x}^* - \mathbf{x}_m^j, \tilde{\mathfrak{g}}_{t,i}(\mathbf{x}_m^j) - \tilde{\mathfrak{g}}_{t,i}(\mathbf{x}_m^i)\rangle}_{:=P_3}\right]
\end{aligned}
\tag{10}
$$

where step (a) is because $f_{t,i}$'s are upper linearizable, step (b) is because $\tilde{\mathfrak{g}}_{t,i}(\cdot)$ is unbiased, step (c) is due to the linearity of conditional expectation, step (d) is due to law of iterated expectation, and step (e) rewrites $(\mathbf{x}^* - \mathbf{x}_m^j)$ as $[(\mathbf{x}^* - \mathbf{x}_m^i) + (\mathbf{x}_m^i - \mathbf{x}_m^j)]$ and $\tilde{\mathfrak{g}}_{t,i}(\mathbf{x}_m^j)$ as $[(\tilde{\mathfrak{g}}_{t,i}(\mathbf{x}_m^j) - \tilde{\mathfrak{g}}_{t,i}(\mathbf{x}_m^i)) + \tilde{\mathfrak{g}}_{t,i}(\mathbf{x}_m^i)]$.

As illustrated in step (e) in Equation 10, total regret can be divided into three parts, $P_1, P_2, P_3$, and we provide upper bound for each of these three parts with proof in the following sections, C.2, C.3, C.4, respectively. In Section C.1, we introduce some auxiliary variables and lemmas that are useful to the following proof.

## C.1 Auxiliary variables and lemmas

In this section, we introduce some auxiliary variables and lemmas to better present the proof of bound for each of the three parts of the total regret presented in Equation 10.

Let $\mathbf{y}_1^i = \tilde{\mathbf{y}}_m^i = \mathbf{c}$, for any $i \in [N]$, since Algorithm 1 would only generate $\mathbf{y}_m^i$ for $m = 2, \cdots, T/K$. Let $\mathbf{r}_m^i = \tilde{\mathbf{y}}_m^i - \mathbf{y}_m^i$ for any $i \in [N]$ and $m \in [T/K]$. For any $m \in [T/K]$, we denote the averages by:

$$
\bar{\mathbf{x}}_m = \frac{\sum_{i=1}^{N} \mathbf{x}_m^i}{N}, \ \bar{\mathbf{y}}_m = \frac{\sum_{i=1}^{N} \mathbf{y}_m^i}{N}, \ \hat{\mathbf{y}}_m = \frac{\sum_{i=1}^{N} \tilde{\mathbf{y}}_m^i}{N}, \ \text{and } \bar{\mathbf{r}}_m = \frac{\sum_{i=1}^{N} \mathbf{r}_m^i}{N}.
$$

There are two lemmas that would be useful to the proofs, and we provide proof of each of these in the following appendices.

**Lemma 9.** *For any $i \in [N]$ and $m \in [T/K]$, Algorithm 1 ensures*

$$
\|\mathbf{r}_m^i\| \leq 2\sqrt{3\epsilon} + 2\eta KG.
$$

*Proof.* See Appendix C.7 for complete proof for Lemma 9. □

**Lemma 10.** *For any $i \in [N]$ and $m \in [T/K]$, Algorithm 1 ensures*

$$\sqrt{\sum_{i=1}^{N} \|\hat{\mathbf{y}}_m - \tilde{\mathbf{y}}_m^i\|^2} \leq \frac{\sqrt{N}(3\eta KG + 2\sqrt{3\epsilon})}{1 - \lambda_2}, \tag{11}$$

$$\sqrt{\sum_{i=1}^{N} \|\hat{\mathbf{y}}_m - \mathbf{y}_{m+1}^i\|^2} \leq \frac{\sqrt{N}(3\eta KG + 2\sqrt{3\epsilon})}{1 - \lambda_2}, \text{ and} \tag{12}$$

$$\sum_{i=1}^{N} \|\mathbf{x}_m^i - \mathbf{x}_m^j\| \leq \left(3\sqrt{2\epsilon} + \frac{(3\eta KG + 2\sqrt{3\epsilon})}{1 - \lambda_2}\right)(N^{3/2} + N). \tag{13}$$

*Proof.* See Appendix C.8 for complete proof of Lemma 10. □

## C.2    Bound of $P_1$

By replacing $(\mathbf{x}^* - \mathbf{x}_m^i)$ with $[(\mathbf{x}^* - \hat{\mathbf{y}}_m) + (\hat{\mathbf{y}}_m - \tilde{\mathbf{y}}_m^i) + (\tilde{\mathbf{y}}_m^i - \mathbf{x}_m^i)]$, $P_1$ can be decomposed as:

$$\begin{aligned}
\frac{1}{\beta}\mathbb{E}[P_1] =& \frac{1}{N}\sum_{i=1}^{N}\sum_{m=1}^{T/K}\sum_{t \in \mathcal{T}_m} \langle \hat{\mathbf{y}}_m - \tilde{\mathbf{y}}_m^i, \tilde{\mathfrak{g}}_{t,i}(\mathbf{x}_m^i)) \rangle + \frac{1}{N}\sum_{i=1}^{N}\sum_{m=1}^{T/K}\sum_{t \in \mathcal{T}_m} \langle \tilde{\mathbf{y}}_m^i - \mathbf{x}_m^i, \tilde{\mathfrak{g}}_{t,i}(\mathbf{x}_m^i) \rangle \\
&+ \underbrace{\frac{1}{N}\sum_{i=1}^{N}\sum_{m=1}^{T/K}\sum_{t \in \mathcal{T}_m} \langle \mathbf{x}^* - \hat{\mathbf{y}}_m, \tilde{\mathfrak{g}}_{t,i}(\mathbf{x}_m^i) \rangle}_{:=P_4} \\
\overset{(a)}{\leq}& \frac{G}{N}\left[\sum_{i=1}^{N}\sum_{m=1}^{T/K}\sum_{t \in \mathcal{T}_m} \|\hat{\mathbf{y}}_m - \tilde{\mathbf{y}}_m^i\| + \sum_{i=1}^{N}\sum_{m=1}^{T/K}\sum_{t \in \mathcal{T}_m} \|\tilde{\mathbf{y}}_m^i - \mathbf{x}_m^i\|\right] + P_4 \\
\overset{(b)}{\leq}& G\left(\sum_{m=1}^{T/K}\sum_{t \in \mathcal{T}_m}\sqrt{\frac{1}{N}\sum_{i=1}^{N}\|\hat{\mathbf{y}}_m - \tilde{\mathbf{y}}_m^i\|^2} + T\sqrt{3\epsilon}\right) + \mathcal{R}_4 \\
\overset{(c)}{\leq}& TG\left(\frac{3\eta KG + 2\sqrt{3\epsilon}}{1 - \lambda_2} + \sqrt{3\epsilon}\right) + \mathcal{R}_4
\end{aligned} \tag{14}$$

where step (a) follows from Cauchy-Schwartz inequality and the bound of $\tilde{\mathfrak{g}}_{t,i}(\cdot)$ function, step (b) follows from Arithmetic Mean-Quadratic Mean inequality and Lemma 1, and step (c) follows from Equation (11) in Lemma 10.

To attain upper bound on $P_4$, we notice that,

$$\begin{aligned}
\bar{\mathbf{y}}_{m+1} = \frac{1}{N}\sum_{i=1}^{N}\mathbf{y}_{m+1}^i =& \frac{1}{N}\sum_{i=1}^{N}\left(\sum_{j \in \mathcal{N}_i} a_{ij}\tilde{\mathbf{y}}_m^j + \eta \sum_{t \in \mathcal{T}_m} \tilde{\mathfrak{g}}_{t,i}(\mathbf{x}_m^i)\right) \\
\overset{(a)}{=}& \frac{1}{N}\sum_{i=1}^{N}\sum_{j=1}^{N} a_{ij}\tilde{\mathbf{y}}_m^j + \frac{\eta}{N}\sum_{i=1}^{N}\sum_{t \in \mathcal{T}_m} \tilde{\mathfrak{g}}_{t,i}(\mathbf{x}_m^i) \\
\overset{(b)}{=}& \hat{\mathbf{y}}_m + \frac{\eta}{N}\sum_{i=1}^{N}\sum_{t \in \mathcal{T}_m} \tilde{\mathfrak{g}}_{t,i}(\mathbf{x}_m^i)
\end{aligned} \tag{15}$$

where step (a) is because $a_{ij} = 0$ for any agent $j \notin \mathcal{N}_i$, and step (b) is because $\mathbf{A1} = \mathbf{1}$.

Substituting $\bar{\mathbf{y}}_{m+1}$ using Equation (14), we have for any $m \in [T/K]$,

$$
\begin{aligned}
\hat{\mathbf{y}}_{m+1} &= \hat{\mathbf{y}}_{m+1} - \bar{\mathbf{y}}_{m+1} + \bar{\mathbf{y}}_{m+1} \\
&\stackrel{(15)}{=} \bar{\mathbf{r}}_{m+1} + \hat{\mathbf{y}}_m + \frac{\eta}{N} \sum_{i=1}^{N} \sum_{t \in \mathcal{T}_m} \tilde{\mathfrak{g}}_{t,i}(\mathbf{x}_m^i)
\end{aligned}
\tag{16}
$$

because $\bar{\mathbf{r}}_{m+1} = \hat{\mathbf{y}}_{m+1} - \bar{\mathbf{y}}_{m+1}$.

By replacing $\hat{\mathbf{y}}_{m+1}$ with Equation (16) and then expand the equation, we have

$$
\begin{aligned}
\|\hat{\mathbf{y}}_{m+1} - \mathbf{x}^*\|^2 &\stackrel{(16)}{=} \left\| \bar{\mathbf{r}}_{m+1} + \hat{\mathbf{y}}_m + \frac{\eta}{N} \sum_{i=1}^{N} \sum_{t \in \mathcal{T}_m} \tilde{\mathfrak{g}}_{t,i}(\mathbf{x}_m^i) - \mathbf{x}^* \right\|^2 \\
&= \|\hat{\mathbf{y}}_m - \mathbf{x}^*\|^2 + 2 \left\langle \hat{\mathbf{y}}_m - \mathbf{x}^*, \frac{\eta}{N} \sum_{i=1}^{N} \sum_{t \in \mathcal{T}_m} \tilde{\mathfrak{g}}_{t,i}(\mathbf{x}_m^i) \right\rangle \\
&\quad + 2 \langle \hat{\mathbf{y}}_m - \mathbf{x}^*, \bar{\mathbf{r}}_{m+1} \rangle + \left\| \bar{\mathbf{r}}_{m+1} + \frac{\eta}{N} \sum_{i=1}^{N} \sum_{t \in \mathcal{T}_m} \tilde{\mathfrak{g}}_{t,i}(\mathbf{x}_m^i) \right\|^2.
\end{aligned}
\tag{17}
$$

Following Lemma 1, we deduce that for any $m \in [T/K]$,

$$
\begin{aligned}
\|\tilde{\mathbf{y}}_{m+1}^i - \mathbf{x}^*\|^2 &\leq \|\mathbf{y}_{m+1}^i - \mathbf{x}^*\|^2 = \|\mathbf{y}_{m+1}^i - \tilde{\mathbf{y}}_{m+1}^i + \tilde{\mathbf{y}}_{m+1}^i - \mathbf{x}^*\|^2 \\
&= \|\mathbf{y}_{m+1}^i - \tilde{\mathbf{y}}_{m+1}^i\|^2 + 2\langle \mathbf{y}_{m+1}^i - \tilde{\mathbf{y}}_{m+1}^i, \tilde{\mathbf{y}}_{m+1}^i - \mathbf{x}^* \rangle + \|\tilde{\mathbf{y}}_{m+1}^i - \mathbf{x}^*\|^2 \\
&= \|\mathbf{r}_{m+1}^i\|^2 - 2\langle \mathbf{r}_{m+1}^i, \tilde{\mathbf{y}}_{m+1}^i - \mathbf{x}^* \rangle + \|\tilde{\mathbf{y}}_{m+1}^i - \mathbf{x}^*\|^2
\end{aligned}
$$

where the last equality is due to the definition of $\mathbf{r}_{m+1}^i$.

Omitting $\|\tilde{\mathbf{y}}_{m+1}^i - \mathbf{x}^*\|^2$ on both sides and moving $\langle \tilde{\mathbf{y}}_{m+1}^i - \mathbf{x}^*, \mathbf{r}_{m+1}^i \rangle$ to the left, we have

$$
\langle \tilde{\mathbf{y}}_{m+1}^i - \mathbf{x}^*, \mathbf{r}_{m+1}^i \rangle \leq \frac{1}{2} \|\mathbf{r}_{m+1}^i\|^2.
\tag{18}
$$

Thus, we can bound the term $\langle \hat{\mathbf{y}}_m - \mathbf{x}^*, \bar{\mathbf{r}}_{m+1} \rangle$ in Equation (17) as follows

$$
\begin{aligned}
\langle \hat{\mathbf{y}}_m - \mathbf{x}^*, \bar{\mathbf{r}}_{m+1} \rangle &= \frac{1}{N} \sum_{i=1}^{N} \langle \hat{\mathbf{y}}_m - \mathbf{x}^*, \mathbf{r}_{m+1}^i \rangle \\
&\stackrel{(a)}{\leq} \frac{1}{N} \sum_{i=1}^{N} \langle \hat{\mathbf{y}}_m - \mathbf{y}_{m+1}^i, \mathbf{r}_{m+1}^i \rangle + \frac{1}{N} \sum_{i=1}^{N} \langle \tilde{\mathbf{y}}_{m+1}^i - \mathbf{x}^*, \mathbf{r}_{m+1}^i \rangle \\
&\stackrel{(b)}{\leq} \frac{1}{N} \sum_{i=1}^{N} \|\hat{\mathbf{y}}_m - \mathbf{y}_{m+1}^i\| \|\mathbf{r}_{m+1}^i\| + \frac{1}{2N} \sum_{i=1}^{N} \|\mathbf{r}_{m+1}^i\|^2 \\
&\stackrel{(c)}{\leq} \frac{2\eta K G + 2\sqrt{3\epsilon}}{\sqrt{N}} \sqrt{\sum_{i=1}^{N} \|\hat{\mathbf{y}}_m - \mathbf{y}_{m+1}^i\|^2} + \frac{1}{2N} \sum_{i=1}^{N} \|\mathbf{r}_{m+1}^i\|^2 \\
&\stackrel{(d)}{\leq} \frac{1}{1 - \lambda_2} \left( 6\eta^2 K^2 G^2 + 10\eta K G \sqrt{3\epsilon} + 12\epsilon \right) \\
&\quad + 2 \left( \eta^2 K^2 G^2 + 2\eta K G \sqrt{3\epsilon} + 3\epsilon \right)
\end{aligned}
\tag{19}
$$

where step (a) replaces $\hat{\mathbf{y}}_m - \mathbf{x}^*$ as $(\hat{\mathbf{y}}_m - \mathbf{y}_{m+1}^i) + (\mathbf{y}_{m+1}^i - \tilde{\mathbf{y}}_{m+1}^i) + (\tilde{\mathbf{y}}_{m+1}^i - \mathbf{x}^*)$ and omits $\langle \mathbf{y}_{m+1}^i - \tilde{\mathbf{y}}_{m+1}^i, \mathbf{r}_{m+1}^i \rangle \leq 0$, step (b) is due to Cauchy-Schwartz inequality and Equation (18), step (c) comes from Lemma 9 and Arithmetic Mean-Quadratic Mean inequality, and step (d) follows from Equation 11 in Lemma 10.

Also, we can bound the last term in Equation (17) as follows

$$
\begin{aligned}
\left\| \bar{\mathbf{r}}_{m+1} + \frac{\eta}{N} \sum_{i=1}^{N} \sum_{t \in \mathcal{T}_m} \tilde{\mathfrak{g}}_{t,i}(\mathbf{x}_m^i) \right\|^2 &\overset{(a)}{\leq} 2\|\bar{\mathbf{r}}_{m+1}\|^2 + 2 \left\| \frac{\eta}{N} \sum_{i=1}^{N} \sum_{t \in \mathcal{T}_m} \tilde{\mathfrak{g}}_{t,i}(\mathbf{x}_m^i) \right\|^2 \\
&\overset{(b)}{\leq} \frac{2}{N} \sum_{i=1}^{N} \|\mathbf{r}_{m+1}^i\|^2 + \frac{2K\eta^2}{N} \sum_{i=1}^{N} \sum_{t \in \mathcal{T}_m} \|\tilde{\mathfrak{g}}_{t,i}(\mathbf{x}_m^i)\|^2 \\
&\overset{(c)}{\leq} 8\left(\sqrt{3\epsilon} + \eta KG\right)^2 + 2\eta^2 K^2 G^2 \\
&= 24\epsilon + 10\eta^2 K^2 G^2 + 16\eta KG\sqrt{3\epsilon}
\end{aligned}
\tag{20}
$$

where both step (a) and step (b) utilize Cauchy-Schwartz inequality and step (c) is due to Lemma 9 and the bound of $\tilde{\mathfrak{g}}_{t,i}(\cdot)$ functions.

Substitute Equation (19) and Equation (20) into Equation (17) and taking expectation on both sides, we have

$$
\begin{aligned}
\mathbb{E}\left[\|\mathbf{x}^* - \hat{\mathbf{y}}_{m+1}\|^2\right] \leq\ & \mathbb{E}\left[\|\mathbf{x}^* - \hat{\mathbf{y}}_m\|^2\right] - \frac{2\eta}{N} \sum_{i=1}^{N} \sum_{t \in \mathcal{T}_m} \mathbb{E}\left[\langle \mathbf{x}^* - \hat{\mathbf{y}}_m, \tilde{\mathfrak{g}}_{t,i}(\mathbf{x}_m^i)\rangle\right] \\
& + \frac{1}{1-\lambda_2}\left(12\eta^2 K^2 G^2 + 20\eta KG\sqrt{3\epsilon} + 24\epsilon\right) \\
& + \left(14\eta^2 K^2 G^2 + 24\eta KG\sqrt{3\epsilon} + 36\epsilon\right).
\end{aligned}
\tag{21}
$$

Moving terms to different sides in Equation (21), we have

$$
\begin{aligned}
\mathbb{E}\left[P_4\right] =\ & \frac{1}{N} \sum_{i=1}^{N} \sum_{m=1}^{T/K} \sum_{t \in \mathcal{T}_m} \mathbb{E}\left[\langle \mathbf{x}^* - \hat{\mathbf{y}}_m, \tilde{\mathfrak{g}}_{t,i}(\mathbf{x}_m^i)\rangle\right] \\
& \overset{(21)}{\leq} \sum_{m=1}^{T/K} \left[\frac{\mathbb{E}\left[\|\hat{\mathbf{y}}_m - \mathbf{x}^*\|^2\right] - \mathbb{E}\left[\|\hat{\mathbf{y}}_{m+1} - \mathbf{x}^*\|^2\right]}{2\eta}\right] \\
& + \frac{T}{K}\left[\frac{18\epsilon}{\eta} + 7\eta K^2 G^2 + 12KG\sqrt{3\epsilon}\right] \\
& + \frac{T}{K}\left[\frac{1}{1-\lambda_2}\left(6\eta K^2 G^2 + 10KG\sqrt{3\epsilon} + \frac{12\epsilon}{\eta}\right)\right] \\
& \overset{(a)}{\leq} \frac{2R^2}{\eta} + \frac{18\epsilon T}{\eta K} + 7\eta TKG^2 + 12TG\sqrt{3\epsilon} \\
& + \frac{1}{1-\lambda_2}\left(\frac{12\epsilon T}{\eta K} + 6\eta TKG^2 + 10TG\sqrt{3\epsilon}\right)
\end{aligned}
\tag{22}
$$

where the last inequality is due to $\mathbb{E}\left[\|\hat{\mathbf{y}}_{T/K+1} - \mathbf{x}^*\|^2\right] \geq 0$ and $\|\hat{\mathbf{y}}_1 - \mathbf{x}^*\|^2 \leq 4R^2$, which is derived by combining $\hat{\mathbf{y}}_1 = \mathbf{c} \in \mathcal{K}$, $\mathbf{x}^* \in \mathcal{K}$, and $R = \max_{\mathbf{x} \in \mathcal{K}} \|\mathbf{x}\|$, and Cauchy-Schwarz Inequality.

Therefore, plugging the result for term $\mathbb{E}\left[P_4\right]$ from Equation (22) into Equation (14), we have

$$
\mathbb{E}\left[P_1\right] \leq TG\beta\left(\frac{3\eta KG + 2\sqrt{3\epsilon}}{1-\lambda_2} + \sqrt{3\epsilon}\right)
\tag{23}
$$

$$
\begin{aligned}
& + \frac{2\beta R^2}{\eta} + \frac{18\beta\epsilon T}{\eta K} + 7\beta\eta TKG^2 + 12TG\beta\sqrt{3\epsilon} \\
& + \frac{\beta}{1-\lambda_2}\left(\frac{12\epsilon T}{\eta K} + 6\eta TKG^2 + 10TG\sqrt{3\epsilon}\right)
\end{aligned}
\tag{24}
$$

### C.3   Bound of $P_2$

Next, we bound $P_2$ in (10)

$$
\begin{aligned}
\mathbb{E}\left[P_2\right] &\overset{(10)}{=} \frac{\beta}{N} \sum_{i=1}^{N} \sum_{m=1}^{T/K} \sum_{t\in\mathcal{T}_m} \langle \mathbf{x}_m^i - \mathbf{x}_m^j, \tilde{\mathfrak{g}}_{t,i}(\mathbf{x}_m^i)\rangle \\
&\overset{(a)}{\le} \frac{\beta}{N} \sum_{i=1}^{N} \sum_{m=1}^{T/K} \sum_{t\in\mathcal{T}_m} \|\mathbf{x}_m^i - \mathbf{x}_m^j\|\|\tilde{\mathfrak{g}}_{t,i}(\mathbf{x}_m^i)\| \\
&\overset{(b)}{\le} \frac{G\beta}{N} \sum_{m=1}^{T/K} \sum_{t\in\mathcal{T}_m} \sum_{i=1}^{N} \|\mathbf{x}_m^i - \mathbf{x}_m^j\| \\
&\overset{(c)}{\le} G\beta(N^{1/2} - +1)\left(3\sqrt{2\epsilon} + \frac{3\eta K G + 2\sqrt{3\epsilon}}{1-\lambda_2}\right).
\end{aligned}
\tag{25}
$$

where step (a) is due to Cauchy-Schwartz inequality, step (b) follows from the bound of $\tilde{\mathfrak{g}}_{t,i}(\cdot)$ functions, and step (c) uses the inequality (13) in Lemma 10.

### C.4   Bound of $P_3$

Recall that $\mathfrak{g}_{t,i}(\cdot)$ are $L_1$-Lipschitz continuous, i.e.,

$$
\|\mathfrak{g}_{t,i}(\mathbf{x}_m^j) - \mathfrak{g}_{t,i}(\mathbf{x}_m^i)\| \le L_1 \|\mathbf{x}_m^i - \mathbf{x}_m^j\|.
$$

Thus, we have

$$
\begin{aligned}
\mathbb{E}\left[P_3\right] &\overset{(10)}{=} \frac{\beta}{N} \sum_{i=1}^{N} \sum_{m=1}^{T/K} \sum_{t\in\mathcal{T}_m} \mathbb{E}\langle \mathbf{x}^* - \mathbf{x}_m^j, \tilde{\mathfrak{g}}_{t,i}(\mathbf{x}_m^j) - \tilde{\mathfrak{g}}_{t,i}(\mathbf{x}_m^i)\rangle \\
&\overset{(a)}{=} \frac{\beta}{N} \sum_{i=1}^{N} \sum_{m=1}^{T/K} \sum_{t\in\mathcal{T}_m} \mathbb{E}\left[\langle \mathbf{x}^* - \mathbf{x}_m^j, \mathbb{E}\left[\tilde{\mathfrak{g}}_{t,i}(\mathbf{x}_m^j)|\mathbf{x}_m^j\right] - \mathbb{E}\left[\tilde{\mathfrak{g}}_{t,i}(\mathbf{x}_m^i)|\mathbf{x}_m^i\right]\rangle\right] \\
&\overset{(b)}{=} \frac{\beta}{N} \sum_{i=1}^{N} \sum_{m=1}^{T/K} \sum_{t\in\mathcal{T}_m} \mathbb{E}\left[\langle \mathbf{x}^* - \mathbf{x}_m^j, \mathfrak{g}_{t,i}(\mathbf{x}_m^j) - \mathfrak{g}_{t,i}(\mathbf{x}_m^i)\rangle\right] \\
&\overset{(c)}{\le} \frac{\beta}{N} \sum_{i=1}^{N} \sum_{m=1}^{T/K} \sum_{t\in\mathcal{T}_m} \mathbb{E}\left[\|\mathbf{x}^* - \mathbf{x}_m^j\|\|\mathfrak{g}_{t,i}(\mathbf{x}_m^j) - \mathfrak{g}_{t,i}(\mathbf{x}_m^i)\|\right] \\
&\overset{(d)}{\le} \frac{L_1\beta}{N} \sum_{i=1}^{N} \sum_{m=1}^{T/K} \sum_{t\in\mathcal{T}_m} \mathbb{E}\left[\|\mathbf{x}^* - \mathbf{x}_m^j\|\|\mathbf{x}_m^i - \mathbf{x}_m^j\|\right] \\
&\overset{(e)}{\le} \frac{2L_1 R\beta}{N} \mathbb{E}\left[\sum_{m=1}^{T/K} \sum_{t\in\mathcal{T}_m} \sum_{i=1}^{N} \|\mathbf{x}_m^i - \mathbf{x}_m^j\|\right] \\
&\overset{(f)}{\le} 2L_1 R\beta(N^{1/2} + 1)\left(3\sqrt{2\epsilon} + \frac{3\eta K G + 2\sqrt{3\epsilon}}{1-\lambda_2}\right)
\end{aligned}
\tag{26}
$$

where step (a) and (b) is due to the law of iterated expectations, step (c) comes from Cauchy-Schwartz inequality, step (d) follows from continuity of $\mathfrak{g}_{t,i}(\cdot)$ functions, step (e) follows from the bound on $\mathcal{K}$ and Cauchy-Schwartz inequality, and step (f) is due to Equation (13) in Lemma 10.

### C.5 Final Regret Bound

Plugging Equation (23), Equation (25) and Equation (26) into Equation (10), for any $j \in [N]$, we have

$$
\begin{aligned}
\mathbb{E}\left[\mathcal{R}_{T,\alpha}^j\right] \leq &\ \frac{2\beta R^2}{\eta} + \frac{18\epsilon\beta T}{\eta K} + 7\eta\beta TKG^2 + 13TG\beta\sqrt{3\epsilon} \\
&+ \frac{\beta}{1-\lambda_2}\left(\frac{12\epsilon T}{\eta K} + 9\eta TKG^2 + 12TG\sqrt{3\epsilon}\right) \\
&+ G\beta(N^{1/2}+1)\left(3\sqrt{2\epsilon} + \frac{(3\eta KG + 2\sqrt{3\epsilon})}{1-\lambda_2}\right) \\
&+ 2L_1 RT\beta(N^{1/2}+1)\left(3\sqrt{2\epsilon} + \frac{(3\eta KG + 2\sqrt{3\epsilon})}{1-\lambda_2}\right)
\end{aligned}
$$

### C.6 Number of Linear Optimization Oracle Calls

Finally, we analyze the total number of linear optimization oracle calls for each agent $i$. In Lemma 1, the term $\mathcal{R}_5 = \left\|\mathbf{y}_{m+1}^i - \sum_{j\in\mathcal{N}_i} a_{ij}\mathbf{x}_m^j\right\|^2$ can be bounded as follows

$$
\begin{aligned}
\mathcal{R}_5 = \left\|\mathbf{y}_{m+1}^i - \sum_{j\in\mathcal{N}_i} a_{ij}\mathbf{x}_m^j\right\|^2 &\overset{(a)}{\leq} 2\left\|\mathbf{y}_{m+1}^i - \sum_{j\in\mathcal{N}_i} a_{ij}\tilde{\mathbf{y}}_m^j\right\|^2 + 2\left\|\sum_{j\in\mathcal{N}_i} a_{ij}\tilde{\mathbf{y}}_m^j - \sum_{j\in\mathcal{N}_i} a_{ij}\mathbf{x}_m^j\right\|^2 \\
&\overset{(b)}{\leq} 2\left\|\eta\sum_{t\in\mathcal{T}_m} \tilde{\mathfrak{g}}_{t,i}(\mathbf{x}_m^i)\right\|^2 + 2\sum_{j\in\mathcal{N}_i} a_{ij}\left\|\tilde{\mathbf{y}}_m^j - \mathbf{x}_m^j\right\|^2 \\
&\overset{(c)}{\leq} 2\eta^2 K^2 G^2 + 6\epsilon
\end{aligned}
\tag{27}
$$

where both step (a) follows from Cauchy-Schwartz inequality, step (b) follows from Cauchy-Schwartz inequality and Line (10) in Algorithm 1

From Equation (3) in Lemma 1, in each block $m$, each agent $i$ in Algorithm 1 at most utilizes

$$
\begin{aligned}
l_m^i &= \frac{27R^2}{\epsilon}\max\left(\frac{\|\mathbf{y}_{m+1}^i - \sum_{j\in\mathcal{N}_i} a_{ij}\mathbf{x}_m^j\|^2(\|\mathbf{y}_{m+1}^i - \sum_{j\in\mathcal{N}_i} a_{ij}\mathbf{x}_m^j\|^2 - \epsilon)}{4\epsilon^2} + 1, 1\right) \\
&= \frac{27R^2}{\epsilon}\max\left(\frac{\mathcal{R}_5(\mathcal{R}_5 - \epsilon)}{4\epsilon^2} + 1, 1\right) \\
&\overset{(27)}{\leq} \frac{27R^2}{\epsilon}\max\left(\frac{(2\eta^2 K^2 G^2 + 6\epsilon)(2\eta^2 K^2 G^2 + 6\epsilon - \epsilon)}{4\epsilon^2} + 1, 1\right) \\
&= \frac{27R^2}{\epsilon}\frac{(2\eta^2 K^2 G^2 + 6\epsilon)(2\eta^2 K^2 G^2 + 5\epsilon) + 4\epsilon^2}{4\epsilon^2}
\end{aligned}
\tag{28}
$$

linear optimization oracle calls, where the last equality is due to the fact that $(2\eta^2 K^2 G^2 + 6\epsilon)(2\eta^2 K^2 G^2 + 5\epsilon) \geq 0$.

Thus, by summing Equation (28) over $\frac{T}{K}$ blocks, we have that the total number of linear optimization steps required by each agent $i$ of Algorithm 1 is at most

$$
\sum_{m=1}^{T/K} l_m^i \leq \frac{27TR^2}{\epsilon K}\left(8.5 + 5.5\frac{K^2\eta^2 G^2}{\epsilon} + \frac{K^4\eta^4 G^4}{\epsilon^2}\right)
$$

## C.7 Proof of Lemma 9

$$
\begin{aligned}
\|\mathbf{r}_{m+1}^i\| = \|\tilde{\mathbf{y}}_{m+1}^i - \mathbf{y}_{m+1}^i\| &\overset{(a)}{\leq} \left\|\tilde{\mathbf{y}}_{m+1}^i - \sum_{j\in\mathcal{N}_i} a_{ij}\mathbf{x}_m^j\right\| + \left\|\sum_{j\in\mathcal{N}_i} a_{ij}\mathbf{x}_m^j - \mathbf{y}_{m+1}^i\right\| \\
&\overset{(b)}{\leq} 2\left\|\sum_{j\in\mathcal{N}_i} a_{ij}\mathbf{x}_m^j - \mathbf{y}_{m+1}^i\right\| \overset{(c)}{=} 2\left\|\sum_{j\in\mathcal{N}_i} a_{ij}\mathbf{x}_m^j - \sum_{j\in\mathcal{N}_i} a_{ij}\tilde{\mathbf{y}}_m^j - \eta \sum_{t=(m-1)K+1}^{mK} \tilde{\mathfrak{g}}_{t,i}(\mathbf{x}_m^i)\right\| \\
&\overset{(d)}{\leq} 2\sum_{j\in\mathcal{N}_i} a_{ij}\|\mathbf{x}_m^j - \tilde{\mathbf{y}}_m^j\| + 2\eta K G \\
&\overset{(e)}{\leq} 2\sqrt{3\epsilon} + 2\eta K G
\end{aligned}
$$

where step (a) comes from triangle inequality, step (b) follows by Lemma 1, step (c) replaces $\mathbf{y}_{m+1}^i$ with update rule described in Algorithm 1, step (d) comes from triangle inequality, and step (e) follows by Lemma 1.

Moreover, if $m = 1$, we can verify that

$$\|\mathbf{r}_1^i\| = \|\mathbf{0}\| \leq 2\sqrt{3\epsilon} + 2\eta K G.$$

## C.8 Proof of Lemma 10

To prove Lemma 10, we introduce additional auxiliary variables as follows:

$$\mathbf{x}_m' = [\mathbf{x}_m^1; \cdots; \mathbf{x}_m^N] \in \mathbb{R}^{Nd}, \ \mathbf{y}_m' = [\mathbf{y}_m^1; \cdots; \mathbf{y}_m^N] \in \mathbb{R}^{Nd}, \ \tilde{\mathbf{y}}_m' = [\tilde{\mathbf{y}}_m^1; \cdots; \tilde{\mathbf{y}}_m^N] \in \mathbb{R}^{Nd}$$

and

$$\mathbf{r}_m' = [\mathbf{r}_m^1; \cdots; \mathbf{r}_m^N] \in \mathbb{R}^{Nd}, \ \mathbf{g}_m' = \sum_{t=(m-1)K+1}^{mK} [\tilde{\mathfrak{g}}_{t,1}(\mathbf{x}_m^1); \cdots; \tilde{\mathfrak{g}}_{t,N}(\mathbf{x}_m^N)] \in \mathbb{R}^{Nd}.$$

According to step 10 in Algorithm 1, for any $m \in \{2, \ldots, T/K\}$, we have

$$\mathbf{y}_{m+1}' = (\mathbf{A} \otimes \mathbf{I})\tilde{\mathbf{y}}_m' + \eta\mathbf{g}_m' = \sum_{k=1}^{m-1} (\mathbf{A} \otimes \mathbf{I})^{m-k}\mathbf{r}_{k+1}' + \sum_{k=1}^{m} (\mathbf{A} \otimes \mathbf{I})^{m-k}\eta\mathbf{g}_k' \tag{29}$$

where the notation $\otimes$ indicates the Kronecker product and $\mathbf{I}$ denotes the identity matrix of size $n \times n$.

In the same manner, for any $m \in [T/K]$, we have

$$
\begin{aligned}
\tilde{\mathbf{y}}_{m+1}' = \mathbf{r}_{m+1}' + \mathbf{y}_{m+1}' &= \mathbf{r}_{m+1}' + (\mathbf{A} \otimes \mathbf{I})\tilde{\mathbf{y}}_m' + \eta\mathbf{g}_m' \\
&= \sum_{k=1}^{m} (\mathbf{A} \otimes \mathbf{I})^{m-k}\mathbf{r}_{k+1}' + \sum_{k=1}^{m} (\mathbf{A} \otimes \mathbf{I})^{m-k}\eta\mathbf{g}_k'
\end{aligned}
\tag{30}
$$

where the second equality follows the fact that $\mathbf{r}_1' = \tilde{\mathbf{y}}_1' - \mathbf{y}_1' = \mathbf{0}$.

By the definition of $\hat{\mathbf{y}}_{m+1}$, for any $m \in [T/K]$, we have

$$[\hat{\mathbf{y}}_{m+1}; \cdots; \hat{\mathbf{y}}_{m+1}] = \left(\frac{\mathbf{1}\mathbf{1}^T}{N} \otimes \mathbf{I}\right) \tilde{\mathbf{y}}_{m+1}' \tag{31}$$

where the second equality comes from $\mathbf{1}^\top \mathbf{A} = \mathbf{1}^\top$.

### C.8.1 Proof of Equation (11)

For any $m \in [T/K]$, we have

$$
\sqrt{\sum_{i=1}^{N} \|\hat{\mathbf{y}}_{m+1} - \tilde{\mathbf{y}}_{m+1}^i\|^2} \overset{(31)}{=} \left\| \left( \frac{\mathbf{1}\mathbf{1}^T}{N} \otimes \mathbf{I} \right) \tilde{\mathbf{y}}_{m+1}' - \tilde{\mathbf{y}}_{m+1}' \right\|
$$

$$
\overset{(30)}{=} \left\| \sum_{k=1}^{m} \left( \left( \frac{\mathbf{1}\mathbf{1}^T}{N} - \mathbf{A}^{m-k} \right) \otimes \mathbf{I} \right) \mathbf{r}_{k+1}' + \sum_{k=1}^{m} \left( \left( \frac{\mathbf{1}\mathbf{1}^T}{N} - \mathbf{A}^{m-k} \right) \otimes \mathbf{I} \right) \eta \mathbf{g}_k' \right\|
$$

$$
\overset{(a)}{\leq} \left\| \sum_{k=1}^{m} \left( \left( \frac{\mathbf{1}\mathbf{1}^T}{N} - \mathbf{A}^{m-k} \right) \otimes \mathbf{I} \right) \mathbf{r}_{k+1}' \right\| + \left\| \sum_{k=1}^{m} \left( \left( \frac{\mathbf{1}\mathbf{1}^T}{N} - \mathbf{A}^{m-k} \right) \otimes \mathbf{I} \right) \eta \mathbf{g}_k' \right\|
$$

$$
\overset{(b)}{\leq} \sum_{k=1}^{m} \left\| \frac{\mathbf{1}\mathbf{1}^T}{N} - \mathbf{A}^{m-k} \right\| \|\mathbf{r}_{k+1}'\| + \sum_{k=1}^{m} \left\| \frac{\mathbf{1}\mathbf{1}^T}{N} - \mathbf{A}^{m-k} \right\| \|\eta \mathbf{g}_k'\|
$$

$$
\overset{(c)}{\leq} \sqrt{N} \sum_{k=1}^{m} \lambda_2^{m-k} (3\eta KG + 2\sqrt{3}\epsilon) \overset{(d)}{\leq} \frac{\sqrt{N}(3\eta KG + 2\sqrt{3}\epsilon)}{1 - \lambda_2},
$$

where step (a) is due to triangle inequality, and step (b) is due to Cauchy-Schwartz inequality and triangle inequality, step (c) comes from Lemma 9 and the fact that $\forall k \in [m], \|\frac{\mathbf{1}\mathbf{1}^T}{N} - \mathbf{A}^{m-k}\| \leq \lambda_2^{m-k}$ (see Mokhtari et al. (2018) for details), and step (d) is due to the property of geometric series.

By noticing $\hat{\mathbf{y}}_1 = \tilde{\mathbf{y}}_1 = \mathbf{c}$, we complete the proof of Equation (11) in Lemma 10.

### C.8.2 Proof of Equation (12)

Similarly, for any $m \in \{2, \cdots, T/K\}$, we have

$$
\sqrt{\sum_{i=1}^{N} \|\hat{\mathbf{y}}_m - \mathbf{y}_{m+1}^i\|^2} \overset{(31)}{=} \left\| \left( \frac{\mathbf{1}\mathbf{1}^T}{N} \otimes \mathbf{I} \right) \tilde{\mathbf{y}}_m' - \mathbf{y}_{m+1}' \right\|
$$

$$
\overset{(29)}{=} \left\| \sum_{k=1}^{m-1} \left( \left( \frac{\mathbf{1}\mathbf{1}^T}{N} - \mathbf{A}^{m-k} \right) \otimes \mathbf{I} \right) \mathbf{r}_{k+1}' + \sum_{k=1}^{m-1} \left( \left( \frac{\mathbf{1}\mathbf{1}^T}{N} - \mathbf{A}^{m-k} \right) \otimes \mathbf{I} \right) \eta \mathbf{g}_k' - \eta \mathbf{g}_m' \right\|
$$

$$
\overset{(a)}{\leq} \left\| \sum_{k=1}^{m-1} \left( \left( \frac{\mathbf{1}\mathbf{1}^T}{N} - \mathbf{A}^{m-k} \right) \otimes \mathbf{I} \right) \mathbf{r}_{k+1}' \right\| + \left\| \sum_{k=1}^{m-1} \left( \left( \frac{\mathbf{1}\mathbf{1}^T}{N} - \mathbf{A}^{m-k} \right) \otimes \mathbf{I} \right) \eta \mathbf{g}_k' \right\| + \|\eta \mathbf{g}_m'\|
$$

$$
\overset{(b)}{\leq} \sum_{k=1}^{m-1} \left\| \frac{\mathbf{1}\mathbf{1}^T}{N} - \mathbf{A}^{m-k} \right\| \|\mathbf{r}_{k+1}'\| + \sum_{k=1}^{m-1} \left\| \frac{\mathbf{1}\mathbf{1}^T}{N} - \mathbf{A}^{m-k} \right\| \|\eta \mathbf{g}_k'\| + \|\eta \mathbf{g}_m'\|
$$

$$
\overset{(c)}{\leq} \sqrt{N} \sum_{k=1}^{m} \lambda_2^{m-k} (3\eta KG + 2\sqrt{3}\epsilon) \overset{(d)}{\leq} \frac{\sqrt{N}(3\eta KG + 2\sqrt{3}\epsilon)}{1 - \lambda_2},
$$

where step (a) is due to triangle inequality, and step (b) is due to Cauchy-Schwartz inequality and triangle inequality, step (c) comes from Lemma 9 and the fact that $\forall k \in [m], \|\frac{\mathbf{1}\mathbf{1}^T}{N} - \mathbf{A}^{m-k}\| \leq \lambda_2^{m-k}$ (see Mokhtari et al. (2018) for details), and step (d) is due to the property of geometric series.

When $m = 1$, $\hat{\mathbf{y}}_1 = \tilde{\mathbf{y}}_1^i = \sum_{j \in \mathcal{N}_i} a_{ij} \tilde{\mathbf{y}}_1^j = \mathbf{c}$. Due to Line (10) in Algorithm 1, we have

$$
\sqrt{\sum_{i=1}^{N} \|\hat{\mathbf{y}}_1 - \mathbf{y}_2^i\|^2} = \sqrt{\sum_{i=1}^{N} \left\| \sum_{j \in \mathcal{N}_i} a_{ij} \tilde{\mathbf{y}}_1^j - \mathbf{y}_2^i \right\|^2} = \sqrt{\sum_{i=1}^{N} \left\| \sum_{t \in \mathcal{T}_m} \tilde{\mathbf{g}}_{t,i}(\mathbf{x}_m^i) \right\|^2} \leq \sqrt{N} \eta KG.
$$

By noticing that $\sqrt{N}\eta KG < \frac{\sqrt{N}(3\eta KG + 2\sqrt{3}\epsilon)}{1-\lambda_2}$, we complete the proof of Equation (12).

### C.8.3 Proof of Equation (13)

For any $m \in [T/K]$, we notice that

$$\sum_{i=1}^{N} \|\mathbf{x}_m^i - \mathbf{x}_m^j\| \le \sum_{i=1}^{N} \|\mathbf{x}_m^i - \bar{\mathbf{x}}_m + \bar{\mathbf{x}}_m - \mathbf{x}_m^j\| \le \sum_{i=1}^{N} \|\mathbf{x}_m^i - \bar{\mathbf{x}}_m\| + N\|\bar{\mathbf{x}}_m - \mathbf{x}_m^j\|$$

$$\le \left(\sqrt{N} + N\right) \sqrt{\sum_{i=1}^{N} \|\mathbf{x}_m^i - \bar{\mathbf{x}}_m\|^2}. \tag{32}$$

Moreover, for any $i \in [N]$ and $m \in \{2, \ldots, T/K\}$, we have

$$\|\bar{\mathbf{x}}_m - \mathbf{x}_m^i\|^2 \le \|\bar{\mathbf{x}}_m - \hat{\mathbf{y}}_m + \hat{\mathbf{y}}_m - \tilde{\mathbf{y}}_m^i + \tilde{\mathbf{y}}_m^i - \mathbf{x}_m^i\|^2$$

$$\overset{(a)}{\le} 3\|\bar{\mathbf{x}}_m - \hat{\mathbf{y}}_m\|^2 + 3\|\hat{\mathbf{y}}_m - \tilde{\mathbf{y}}_m^i\|^2 + 3\|\tilde{\mathbf{y}}_m^i - \mathbf{x}_m^i\|^2$$

$$\overset{(b)}{\le} \frac{3}{N} \sum_{j=1}^{N} \|\mathbf{x}_m^j - \tilde{\mathbf{y}}_m^j\|^2 + 3\|\hat{\mathbf{y}}_m - \tilde{\mathbf{y}}_m^i\|^2 + 3\|\tilde{\mathbf{y}}_m^i - \mathbf{x}_m^i\|^2$$

$$\overset{(c)}{\le} 18\epsilon + 3\|\hat{\mathbf{y}}_m - \tilde{\mathbf{y}}_m^i\|^2,$$

where step (a) utilizes Cauchy-Schwarz inequality, step (b) utilizes Cauchy-Schwarz inequality and the definition of $\hat{\mathbf{y}}_m$ and $\bar{\mathbf{x}}_m$, and step (c) comes from Lemma 1. When $m = 1$, $\|\bar{\mathbf{x}}_1 - \mathbf{x}_1^i\|^2 = \mathbf{0} \le 18\epsilon + 3\|\hat{\mathbf{y}}_m - \tilde{\mathbf{y}}_m^i\|^2$, which leads us to conclude that for any $m \in [T/K]$,

$$\|\bar{\mathbf{x}}_m - \mathbf{x}_m^i\|^2 \le 18\epsilon + 3\|\hat{\mathbf{y}}_m - \tilde{\mathbf{y}}_m^i\|^2.$$

Thus, for any $m \in [T/K]$, we have

$$\sqrt{\sum_{i=1}^{N} \|\bar{\mathbf{x}}_m - \mathbf{x}_m^i\|^2} \le \sqrt{\sum_{i=1}^{N} (18\epsilon + 3\|\hat{\mathbf{y}}_m - \tilde{\mathbf{y}}_m^i\|^2)}$$

$$\overset{(a)}{\le} 3\sqrt{2N\epsilon} + \sqrt{3 \sum_{i=1}^{N} \|\hat{\mathbf{y}}_m - \tilde{\mathbf{y}}_m^i\|^2}$$

$$\overset{(b)}{\le} 3\sqrt{2N\epsilon} + \frac{\sqrt{3N}(3\eta KG + 2\sqrt{3\epsilon})}{1 - \lambda_2} \tag{33}$$

where step (a) is due to triangle inequality and step (b) follows by Equation (11) in Lemma 10.

Finally, by substituting Equation (33) into Equation (32), for any $m \in [T/K]$, we have

$$\sum_{i=1}^{N} \|\mathbf{x}_m^i - \mathbf{x}_m^j\| \le (\sqrt{N} + N) \sqrt{\sum_{i=1}^{N} \|\mathbf{x}_m^i - \bar{\mathbf{x}}_m\|^2}$$

$$\le \left(3\sqrt{2\epsilon} + \frac{(3\eta KG + 2\sqrt{3\epsilon})}{1 - \lambda_2}\right) (N^{3/2} + N).$$

## D  Bandit Feedback for Trivial Query Functions

In this section, we describe and discuss the variation of DOCLO to handle bandit feedback for functions with trivial query oracle. The detailed implementation is given in the Algorithm 2, and here we provide proof of Theorem 3. This algorithm requires additional input from the user: smoothing parameter $\delta \le \alpha$, shrunk set $\hat{\mathcal{K}}_\delta$, linear space $\mathcal{L}_0$.

*Proof.* Let $\mathcal{A}'$ denote Algorithm 2 and $\mathcal{A}$ denote Algorithm 1. Let $\hat{f}_{t,j}$ denote a $\delta$-smoothed version of $f_{t,i}$. Let $\mathbf{x}^* \in \text{argmax}_{\mathbf{x}\in\mathcal{K}} \sum_{t=1}^{T} \sum_{j=1}^{N} f_{t,j}(\mathbf{x})$ and $\hat{\mathbf{x}}^* \in \text{argmax}_{\mathbf{x}\in\hat{\mathcal{K}}_\delta} \sum_{t=1}^{T} \sum_{j=1}^{N} \hat{f}_{t,j}(\mathbf{x})$. Following our description in Section 5.1, Algorithm 2 is equivalent to running Algorithm 1 on $\hat{f}_{t,i}$, a $\delta$ smoothed version of $f_{t,i}$ over a shrunk set $\hat{\mathcal{K}}_\delta$.

By the definition of regret, we have

$$
\begin{aligned}
\mathbb{E}\left[\mathcal{R}_\alpha^{i,\mathcal{A}'}\right] - \mathbb{E}\left[\mathcal{R}_\alpha^{i,\mathcal{A}}\right] &= \frac{1}{N}\mathbb{E}\left[\alpha \sum_{t=1}^{T}\sum_{j=1}^{N} f_{t,j}(\mathbf{x}^*) - \sum_{t=1}^{T}\sum_{j=1}^{N} f_{t,j}(h(\mathbf{x}_t^i) + \delta\mathbf{v}_t^i)\right] \\
&\quad - \frac{1}{N}\mathbb{E}\left[\alpha \sum_{t=1}^{T}\sum_{j=1}^{N} \hat{f}_{t,j}(\hat{\mathbf{x}}^*) - \sum_{t=1}^{T}\sum_{j=1}^{N} \hat{f}_{t,j}(h(\mathbf{x}_t^i))\right] \\
&= \frac{1}{N}\mathbb{E}\left[\left(\sum_{t=1}^{T}\sum_{j=1}^{N} \hat{f}_{t,j}(h(\mathbf{x}_t^i)) - \sum_{t=1}^{T}\sum_{j=1}^{N} f_{t,j}(h(\mathbf{x}_t^i) + \delta\mathbf{v}_t^i)\right)\right. \\
&\quad \left. + \alpha\left(\sum_{t=1}^{T}\sum_{j=1}^{N} f_{t,j}(\mathbf{x}^*) - \sum_{t=1}^{T}\sum_{j=1}^{N} \hat{f}_{t,j}(\hat{\mathbf{x}}^*)\right)\right].
\end{aligned}
\tag{34}
$$

Based on Lemma 3 proved by Pedramfar et al. (2023), we have $|\hat{f}_{t,j}(h(\mathbf{x}_t^i)) - f_{t,j}(h(\mathbf{x}_t^i))| \leq \delta M_1$, and $\hat{f}_{t,j}$ is $M_1$-Lipschitz continuous as well. Thus, we have

$$
|f_{t,j}(h(\mathbf{x}_t^i) + \delta\mathbf{v}_t^i) - \hat{f}_{t,j}(h(\mathbf{x}_t^i))| \leq |f_{t,j}(h(\mathbf{x}_t^i) + \delta\mathbf{v}_t^i) - f_{t,j}(h(\mathbf{x}_t^i))| + |f_{t,j}(h(\mathbf{x}_t^i)) - \hat{f}_{t,j}(h(\mathbf{x}_t^i))| \leq 2\delta M_1.
\tag{35}
$$

Meanwhile, for the second part of Equation 34, we have

$$
\begin{aligned}
\sum_{t=1}^{T}\sum_{j=1}^{N} \hat{f}_{t,j}(\hat{\mathbf{x}}^*) &= \max_{\hat{\mathbf{x}}\in\hat{\mathcal{K}}_\alpha} \sum_{t=1}^{T}\sum_{j=1}^{N} \hat{f}_{t,j}(\hat{\mathbf{x}}) \\
&\overset{(a)}{\geq} -N\delta M_1 T + \max_{\hat{\mathbf{x}}\in\hat{\mathcal{K}}_\alpha} \sum_{t=1}^{T}\sum_{j=1}^{N} f_{t,j}(\hat{\mathbf{x}}) \\
&\overset{(b)}{=} -N\delta M_1 T + \max_{\mathbf{x}\in\mathcal{K}} \sum_{t=1}^{T}\sum_{j=1}^{N} f_{t,j}\left(\left(1-\frac{\delta}{r}\right)\mathbf{x} + \frac{\delta}{r}\mathbf{c}\right) \\
&= -N\delta M_1 T + \max_{\mathbf{x}\in\mathcal{K}} \sum_{t=1}^{T}\sum_{j=1}^{N} f_{t,j}\left(\mathbf{x} + \frac{\delta}{r}(\mathbf{c}-\mathbf{x})\right) \\
&\overset{(c)}{\geq} -N\delta M_1 T + \max_{\mathbf{x}\in\mathcal{K}} \sum_{t=1}^{T}\sum_{j=1}^{N} \left(f_{t,j}(\mathbf{x}) - \frac{4\delta M_1 R}{r}\right) \\
&= -\left(1+\frac{4R}{r}\right)N\delta M_1 T + \sum_{t=1}^{T}\sum_{j=1}^{N} f_{t,j}(\mathbf{x}^*)
\end{aligned}
$$

where step (a) follows from Lemma 3 by Pedramfar et al. (2023), step (b) follows from the definition of $\hat{\mathcal{K}}_\delta$, and step (c) is due to the $M_1$-Lipschitz continuity of $f_{t,i}$'s.

Putting it together with Equation 34 and Equation 35, we have

$$
\mathcal{R}_\alpha^{i,\mathcal{A}'} - \mathcal{R}_\alpha^{i,\mathcal{A}} \leq \frac{1}{N}\left(2N\delta M_1 T + \left(1+\frac{4R}{r}\right)N\delta M_1 T\right) = \left(3+\frac{4R}{r}\right)\delta M_1 T.
$$

Thus we have

$$\mathcal{R}_{\alpha}^{i,\mathcal{A}'} \leq \mathcal{R}_{\alpha}^{i,\mathcal{A}} + \left(3 + \frac{4R}{r}\right)\delta M_1 T.$$

Assuming the zeroth order oracle is bounded by $B_0$, from Line 10 of Algorithm 2 we see that the gradient sample that is being passed to $\mathcal{A}$ is bounded by $G = \frac{k}{\delta}B_0 = O(\delta^{-1})$. Substituting results from Theorem 1, we see that

$$\mathbb{E}\left[\mathcal{R}_{\alpha}^{i,\mathcal{A}'}\right] = O\left(\mathcal{R}_{\alpha}^{i,\mathcal{A}} + \delta T\right) = O\left(\frac{1}{\eta} + \eta T K \delta^{-2} + \delta T\right).$$

Since we are doing same amount of infeasible projection operation and communication operation in Algorithm 2, LOO calls and communication complexity for Algorithm 2 remains the same as Algorithm 1. □

## E    Semi-Bandit Feedback for Non-Trivial Query Functions

To transform DOCLO into an algorithm that can handle semi-bandit feedback when we are dealing with functions with non-trivial queries, we pass $\frac{T}{L}$ as time horizon to DOCLO. In each of those $\frac{T}{L}$ blocks, we consider functions $\left(\hat{f}_{q,i}\right)_{1 \leq q \leq T/L, 1 \leq i \leq N}$, where $\hat{f}_{q,i} = \frac{1}{L}\sum_{t=(q-1)L+1}^{qL} f_{t,i}$. We note that in Algorithm 3, for any $\mathbf{x} \in \mathcal{K}$ and $1 \leq q \leq T/L$, we have $\mathbb{E}[f_{t'_q}(\mathbf{x})] = \hat{f}_q(\mathbf{x})$, and if $f_t$ are differentiable, $\mathbb{E}[\nabla f_{t'_q}(\mathbf{x})] = \nabla \hat{f}_q(\mathbf{x})$. This way, the transformed algorithm only queries once at the point of action per block, thus semi-bandit.

*Proof of Theorem 4.* Let $\mathcal{A}'$ denote Algorithm 3 and $\mathcal{A}$ denote Algorithm 1. Following our description in Section 5.2.1, Algorithm 3 is equivalent to running Algorithm 1 on $\hat{f}_{q,i}(\mathbf{x}) = \frac{1}{L}\sum_{t=(q-1)L+1}^{qL} f_{t,i}(\mathbf{x})$, an average of $f_{t,i}$ over block $q$. In consistence with Algorithm 3 description, we let $\mathbf{z}_t^i$ denote the action taken by agent $i$ at time-step $t$, whether it be $\hat{\mathbf{x}}_q$, point of action selected by DOCLO, or $\hat{\mathbf{y}}_q$, point of query selected by DOCLO. Thus, the regret of Algorithm 3 over horizon $T$ is

$$\begin{aligned}
\mathbb{E}\left[\mathcal{R}_{\alpha,T}^{i,\mathcal{A}'}\right] &= \frac{1}{N}\mathbb{E}\left[\alpha \max_{\mathbf{u}\in\mathcal{K}}\sum_{t=1}^{T}\sum_{j=1}^{N} f_{t,j}(\mathbf{u}) - \sum_{t=1}^{T}\sum_{j=1}^{N} f_{t,j}(\mathbf{z}_t^i)\right] \\
&= \frac{L}{N}\mathbb{E}\left[\alpha \max_{\mathbf{u}\in\mathcal{K}}\frac{1}{L}\sum_{t=1}^{T}\sum_{j=1}^{N} f_{t,j}(\mathbf{u}) - \frac{1}{L}\sum_{t=1}^{T}\sum_{j=1}^{N} f_{t,j}(\mathbf{z}_t^i)\right] \\
&= \frac{L}{N}\mathbb{E}\left[\alpha \max_{\mathbf{u}\in\mathcal{K}}\frac{1}{L}\sum_{j=1}^{N}\sum_{q=1}^{T/L}\sum_{t=(q-1)L+1}^{qL} f_{t,j}(\mathbf{u}) - \frac{1}{L}\sum_{j=1}^{N}\sum_{q=1}^{T/L}\sum_{t=(q-1)L+1}^{qL} f_{t,j}(\mathbf{z}_t^i)\right] \\
&= \frac{1}{N}\mathbb{E}\left[\sum_{j=1}^{N}\sum_{q=1}^{T/L}\sum_{t=(q-1)L+1}^{qL}\left(f_{t,j}(\hat{\mathbf{x}}_q^i) - f_{t,j}(\mathbf{z}_t^i)\right) + L\left(\alpha \max_{\mathbf{u}\in\mathcal{K}}\sum_{j=1}^{N}\sum_{q=1}^{T/L}\hat{f}_{q,j}(\mathbf{u}) - \sum_{j=1}^{N}\sum_{q=1}^{T/L}\hat{f}_{q,j}(\hat{\mathbf{x}}_q^i)\right)\right]
\end{aligned}$$
$$\tag{36}$$

Algorithm 3 ensures that in each block with a given $q$, there is only 1 iteration where $\mathbf{z}_t^i \neq \hat{\mathbf{x}}_q^i$, otherwise $\mathbf{z}_t^i = \hat{\mathbf{x}}_q^i$. Since $f_{t,j}$ are $M_1$-Lipschitz continuous, we have $|f_{t,j}(\hat{\mathbf{x}}_q^i) - f_{t,j}(\hat{\mathbf{y}}_t^i)| \leq M_1|\hat{\mathbf{x}}_q^i - \hat{\mathbf{y}}_t^i| \leq 2M_1 R$ where the second equation comes from the restraint on $\mathcal{K}$. Since $\hat{\mathcal{K}}_\delta \subseteq \mathcal{K}$, we have $\max_{\mathbf{x}\in\hat{\mathcal{K}}_\delta}\|\mathbf{x}\| \leq \max_{\mathbf{x}\in\mathcal{K}}\|\mathbf{x}\| = R$. Thus, we have

$$\sum_{j=1}^{N}\sum_{q=1}^{T/L}\sum_{t=(q-1)L+1}^{qL}\left|f_{t,j}(\hat{\mathbf{x}}_q^i) - f_{t,j}(\mathbf{x}_t^i)\right| \leq \sum_{j=1}^{N}\sum_{q=1}^{T/L}\left(0*(L-1) + 2M_1 R*1\right) = \frac{2NTM_1 R}{L} \tag{37}$$

The second part of Equation 36 can be seen as the regret of running Algorithm 1 against $\left(\hat{f}_{q,i}\right)_{1 \leq q \leq T/L, 1 \leq i \leq N}$, over horizon $T/L$ instead of $T$. We denote it with $\mathcal{R}_{\alpha,T/L}^{i,\mathcal{A}}$. Applying Theorem 1, we have

$$\mathbb{E}\left[R_{\alpha,T/L}^{i,\mathcal{A}}\right] = O\left(\frac{1}{\eta} + \frac{\eta T K G^2}{L}\right).$$

Putting together with Equation 36 and Equation 37, we have

$$\mathbb{E}\left[\mathcal{R}_{\alpha,T}^{i,\mathcal{A}'}\right] \leq \frac{2TM_1 R}{L} + L\mathbb{E}\left[\mathcal{R}_{\alpha,T/L}^{i,\mathcal{A}}\right]$$

which means

$$\mathbb{E}\left[\mathcal{R}_{\alpha}^i\right] = O\left(\frac{L}{\eta} + \eta T K G^2 + \frac{T}{L}\right).$$

Based on the implementation described in Algorithm 3, it queries oracle every $L$ iterations, and communicate and make updates with infeasible projection operation every $KL$ iteration. Thus, communication complexity for Algorithm 3 is $O(\frac{T}{KL})$, while LOO calls are $O(\frac{T}{\epsilon KL})$.

$\square$

# F  Zeroth Order Full-Information Feedback for Non-Trivial Query Functions

In this section, we describe and discuss the variation of DOCLO to handle zeroth-order full-information feedback for functions with non-trivial query oracle. The detailed implementation is given in the Algorithm 4 table, followed by proof of Theorem 5. Per request of FOTZO, Algorithm 4 requires additional input from the user: smoothing parameter $\delta \leq \alpha$, shrunk set $\hat{\mathcal{K}}_\delta$, linear space $\mathcal{L}_0$. In the case of non-trivial query oracle, $h(\cdot)$ is not necessarily an identity function. In the following, we give proof of Theorem 5.

*Proof of Theorem 5.* Let $\mathcal{A}'$ denote Algorithm 4 and $\mathcal{A}$ denote Algorithm 1. Let $\hat{f}_{t,j}$ denote a $\delta$-smoothed version of $f_{t,i}$. Let $\mathbf{x}^* \in \operatorname{argmax}_{\mathbf{x} \in \mathcal{K}} \sum_{t=1}^T \sum_{j=1}^N f_{t,j}(\mathbf{x})$ and $\hat{\mathbf{x}}^* \in \operatorname{argmax}_{\mathbf{x} \in \hat{\mathcal{K}}_\delta} \sum_{t=1}^T \sum_{j=1}^N \hat{f}_{t,j}(\mathbf{x})$. Following our description in Section 5.2.2, Algorithm 4 is equivalent to running Algorithm 1 on $\hat{f}_{t,i}$, a $\delta$ smoothed version of $f_{t,i}$ over a shrunk set $\hat{\mathcal{K}}_\delta$.

By the definition of regret, we have

$$\begin{aligned}
\mathbb{E}\left[\mathcal{R}_\alpha^{i,\mathcal{A}'}\right] - \mathbb{E}\left[\mathcal{R}_\alpha^{i,\mathcal{A}}\right] &= \frac{1}{N}\mathbb{E}\left[\alpha \sum_{t=1}^T \sum_{j=1}^N f_{t,j}(\mathbf{x}^*) - \sum_{t=1}^T \sum_{j=1}^N f_{t,j}(h(\mathbf{x}_t^i))\right] \\
&\quad - \frac{1}{N}\mathbb{E}\left[\alpha \sum_{t=1}^T \sum_{j=1}^N \hat{f}_{t,j}(\hat{\mathbf{x}}^*) - \sum_{t=1}^T \sum_{j=1}^N \hat{f}_{t,j}(h(\mathbf{x}_t^i))\right] \\
&= \frac{1}{N}\mathbb{E}\left[\left(\sum_{t=1}^T \sum_{j=1}^N \hat{f}_{t,j}(h(\mathbf{x}_t^i)) - \sum_{t=1}^T \sum_{j=1}^N f_{t,j}(h(\mathbf{x}_t^i))\right) \right. \\
&\quad \left. +\alpha\left(\sum_{t=1}^T \sum_{j=1}^N f_{t,j}(\mathbf{x}^*) - \sum_{t=1}^T \sum_{j=1}^N \hat{f}_{t,j}(\hat{\mathbf{x}}^*)\right)\right].
\end{aligned} \tag{38}$$

Based on Lemma 3 proved by Pedramfar et al. (2023), we have $|\hat{f}_{t,j}(h(\mathbf{x}_t^i)) - f_{t,j}(h(\mathbf{x}_t^i))| \leq \delta M_1 < 2\delta M_1$.

It can be shown that the second part of Equation 38 follows the same upper bound as Equation 35.

Thus, putting it together, we have for Algorithm 2, the regret

$$\mathcal{R}_\alpha^{i,\mathcal{A}'} \leq \mathcal{R}_\alpha^{i,\mathcal{A}} + \frac{1}{N}\left(2N\delta M_1 T + \left(1 + \frac{4R}{r}\right)N\delta M_1 T\right) = \mathcal{R}_\alpha^{i,\mathcal{A}} + \left(3 + \frac{4R}{r}\right)\delta M_1 T$$

Assuming the zeroth order oracle is bounded by $B_0$, from Line 10 of Algorithm 4 we see that the gradient sample that is being passed to $\mathcal{A}$ is bounded by $G = \frac{k}{\delta}B_0 = O(\delta^{-1})$. Substituting results from Theorem 1, we see that

$$\mathbb{E}\left[\mathcal{R}_\alpha^i\right] = O\left(\frac{1}{\eta} + \eta T K \delta^{-2} + \delta T\right).$$

Since we are doing same amount of infeasible projection operation and communication operation in Algorithm 4, LOO calls and communication complexity for Algorithm 4 remains the same as Algorithm 1. □

## G Bandit Feedback for Non-trivial Query Function

In this section, we extend DOCLO over functions with nontrivial query oracle to handle bandit feedback, i.e., trivial query and zero-order feedback. We achieve this by applying FOTZO to handle zero-order full-information feedback, then applying SFTT to transform the algorithm into trivial queries. In the following, we provide proof of Theorem 6.

*Proof of Theorem 6.* Let $\mathcal{A}'$ denote Algorithm 5 and $\mathcal{A}$ denote Algorithm 4. Following our description in Section 5.2.3, Algorithm 5 is equivalent to running Algorithm 4 on $\hat{f}_{q,i}(\mathbf{x}) = \frac{1}{L}\sum_{t=(q-1)L+1}^{qL}\hat{f}_{t,i}(\mathbf{x})$, an average of $\hat{f}_{t,i}$ over block $q$, where $\hat{f}_{t,i}$ is a $\delta$-smoothed version of $f_{t,i}$. In consistence with Algorithm 5 description, we let $\mathbf{x}_t^i$ denote the action taken by agent $i$ at iteration $t$, whether it be $\hat{\mathbf{x}}_q$, point of action selected by DOCLO, or $\hat{\mathbf{y}}_q$, point of query selected by DOCLO. Thus, the regret of Algorithm 5 over horizon T:

$$\begin{aligned}
\mathbb{E}\left[\mathcal{R}_\alpha^{i,\mathcal{A}'}\right] &= \frac{1}{N}\mathbb{E}\left[\alpha\max_{\mathbf{u}\in\mathcal{K}}\sum_{t=1}^T\sum_{j=1}^N \hat{f}_{t,j}(\mathbf{u}) - \sum_{t=1}^T\sum_{j=1}^N \hat{f}_{t,j}(\mathbf{z}_t^i)\right]\\
&= \frac{L}{N}\mathbb{E}\left[\alpha\max_{\mathbf{u}\in\mathcal{K}}\frac{1}{L}\sum_{t=1}^T\sum_{j=1}^N \hat{f}_{t,j}(\mathbf{u}) - \frac{1}{L}\sum_{t=1}^T\sum_{j=1}^N \hat{f}_{t,j}(\mathbf{z}_t^i)\right]\\
&= \frac{L}{N}\mathbb{E}\left[\alpha\max_{\mathbf{u}\in\mathcal{K}}\frac{1}{L}\sum_{j=1}^N\sum_{q=1}^{T/L}\sum_{t=(q-1)L+1}^{qL} \hat{f}_{t,j}(\mathbf{u}) - \frac{1}{L}\sum_{j=1}^N\sum_{q=1}^{T/L}\sum_{t=(q-1)L+1}^{qL} \hat{f}_{t,j}(\mathbf{z}_t^i)\right]\\
&= \frac{1}{N}\mathbb{E}\Bigg[\sum_{j=1}^N\sum_{q=1}^{T/L}\sum_{t=(q-1)L+1}^{qL}\left(\hat{f}_{t,j}(\hat{\mathbf{x}}_q^i) - \hat{f}_{t,j}(\mathbf{z}_t^i)\right)\\
&\qquad + L\left(\alpha\max_{\mathbf{u}\in\mathcal{K}}\sum_{j=1}^N\sum_{q=1}^{T/L}\hat{f}_{q,j}(\mathbf{u}) - \sum_{j=1}^N\sum_{q=1}^{T/L}\hat{f}_{q,j}(\hat{\mathbf{x}}_q^i)\right)\Bigg]
\end{aligned}\tag{39}$$

Algorithm 3 ensures that in each block $q$, there is only 1 iteration where $\mathbf{z}_t^i = \hat{\mathbf{y}}_t^i \neq \hat{\mathbf{x}}_t^i$, otherwise $\mathbf{x}_t^i = \hat{\mathbf{x}}_t^i$. Based on Lemma 3 proposed by Pedramfar et al. (2023), $\hat{f}_{t,j}$ is $M_1$-Lipscitz continuous if $f_{t,i}$ is $M_1$-Lipscitz continuous, i.e., $|\hat{f}_{t,j}(\hat{\mathbf{x}}_q^i) - \hat{f}_{t,j}(\hat{\mathbf{y}}_t^i)| \leq M_1|\hat{\mathbf{x}}_q^i - \hat{\mathbf{y}}_t^i| \leq 2M_1 R$ where the second equation comes from the restraint on $\mathcal{K}$. Since $\hat{\mathcal{K}}_\delta \subseteq \mathcal{K}$, we have $\max_{\mathbf{x}\in\hat{\mathcal{K}}_\delta}\|\mathbf{x}\| \leq \max_{\mathbf{x}\in\mathcal{K}}\|\mathbf{x}\| = R$. Thus, we have

$$\sum_{j=1}^N\sum_{q=1}^{T/L}\sum_{t=(q-1)L+1}^{qL}\left(\hat{f}_{t,j}(\hat{\mathbf{x}}_q^i) - \hat{f}_{t,j}(\mathbf{x}_t^i)\right) = \sum_{j=1}^N\sum_{q=1}^{T/L}(0*(L-1) + 2M_1 R*1) = \frac{2NTM_1 R}{L}\tag{40}$$

The second part of Equation 39 can be seen as the regret of running Algorithm 4 against $\left(\hat{f}_{q,i}\right)_{1 \leq q \leq T/L, 1 \leq i \leq N}$, over horizon $T/L$ instead of $T$. We denote it with $\mathcal{R}_{\alpha, T/L}^{i, \mathcal{A}}$. Applying Theorem 5, we have

$$\mathbb{E}\left[R_{\alpha, T/L}^{i, \mathcal{A}}\right] = O\left(\frac{1}{\eta} + \frac{\eta T K \delta^{-2}}{L} + \frac{\delta T}{L}\right).$$

Putting it together with Equation 36 and Equation 37, we have

$$\mathbb{E}\left[\mathcal{R}_{\alpha}^{i, \mathcal{A}'}\right] \leq \frac{2 T M_1 R}{L} + L \mathbb{E}\left[\mathcal{R}_{\alpha, T/L}^{i, \mathcal{A}}\right],$$

which means that

$$\mathbb{E}\left[\mathcal{R}_{\alpha}^{i}\right] = O\left(\frac{L}{\eta} + \eta T K \delta^{-2} + \delta T + \frac{T}{L}\right).$$

Based on the implementation described in Algorithm 3, it queries oracle every $L$ iterations, and communicate and make updates with infeasible projection operation every $KL$ iteration. Thus, communication complexity for Algorithm 3 is $O(\frac{T}{KL})$, while LOO calls are $O(\frac{T}{\epsilon KL})$. $\qquad \square$

