# OpenReview forum: "Decentralized Projection-free Online Upper-Linearizable Optimization with Applications to DR-Submodular Optimization"
_TMLR — Accepted by TMLR_

### Review · Reviewer_1K9M · 2025-08-19

**Summary Of Contributions:**

Recently, Pedramfar and Aggrawal (2024) introduced upper linearizable functions which generalizes a collection of functions including DR-submodular and concave functions. This paper, extends this work by introducing decentralized online algorithms for upper linearizable functions which are additionally more efficient by being projection-free. In particular, the algorithm involves a parameter $\theta$ that enables the trade-off between regret vs communication complexity and calls to a linear optimization oracle. Extensions are also shown to settings with different feedback types.

**Audience:**

Yes

**Audience Explanation:**

I think the problem was not solved before and that the technical content is dense enough.

**Broader Impact Concerns:**

I don't think that there are any ethical concerns for this paper.

**Claims And Evidence:**

Yes

**Claims Explanation:**

The paper provides mathematical proofs for the theorems.

**Requested Changes:**

-I don't specialize in this area and did not find this paper easy to follow. Additionally, the paper's main contribution is technical and calibrating for the level of novelty requires good background in the existing literature. The paper has many technical definitions/terminology, e..g., up-concave, \gamma-weakly monotone. These definitions are given only later. Even in the abstract, it was not clear what DR means.

-The algorithm makes call to linear optimization oracles, but aren't these calls expensive?

-are simulation results important in this area? Would the paper benefit by showing empirical results that prove the performance of the algorithm? I did not see any included.

-the action set is introduced before assumption 2 on page 5, but what is the action set?

-were projection-based algorithms considered? Is there a theoretical or empirical result that makes it clear that projection-free algorithms would always be faster?

---

> ### Author Response · Authors · 2025-09-23
> **Response to Reviewer 1K9M (1)**
>
> Many thanks to the reviewer for taking the time to review our paper.
>
> > `I don't specialize in this area and did not find this paper easy to follow. Additionally, the paper's main contribution is technical and calibrating for the level of novelty requires good background in the existing literature. The paper has many technical definitions/terminology, e..g., up-concave, \gamma-weakly monotone. These definitions are given only later. Even in the abstract, it was not clear what DR means.'
>
> We thank the reviewer for this valuable feedback. We agree that the paper's technical density can be a barrier, and we have revised the manuscript to improve clarity for a broader audience. Specifically, we have defined key terms like 'DR' upon their first appearance in the abstract, and added intuitive explanations for specialized concepts like $\gamma$-weakly up-concave functions in the beginning of Introduction section before summarizing our contributions and technical novelties, with reference to detailed and rigorous definitions in the later section Preliminaries.
>
> > `The algorithm makes call to linear optimization oracles, but aren't these calls expensive?'
>
> Thank you for raising the point regarding the computational cost and motivation for using projection-free methods.
> LOO calls are not without cost, our core argument is that the efficiency of LOOs is relative to the often-prohibitive cost of projection oracles, especially in high-dimensional settings.
> A projection oracle would require solving a quadratic optimization problem:
> $\mathcal{O}P(\mathcal{K},\mathbb{x}):=\arg\min_{\mathbb{u} \in \mathcal{K}} \Vert \mathbb{x} - \mathbb{u} \Vert^2$ , while a linear optimization oracle needs only solving a linear program:
> $\mathcal{O}LO(\mathcal{K},\mathbb{x}):=\arg\min_{\mathbb{u} \in \mathcal{K}} \langle \mathbb{x}, \mathbb{u} \rangle$.
> We refer to Chapter 7 in [1] for a detailed discussion on the motivation of projection-free algorithms for online optimizations, and Table 1.1 in [2] for examples of the computational cost of Projection and Linear programs on different feasible sets.
>
> We have updated the second paragraph in Introduction to clarify that:
> - A projection oracle must solve a quadratic optimization problem, which can be computationally expensive for complex constraint sets. In contrast, an LOO solves a more efficient linear programming problem.
> - We also added the references [1,2] to the main text.
> - We explicitly highlight the example of optimization over the nuclear norm ball, where projection requires a full, costly singular value decomposition (SVD), while the LOO only requires finding the top singular vector pair, which is significantly faster.
>
>
> [1] Hazan, Elad. "Introduction to online convex optimization." Foundations and Trends® in Optimization 2.3-4 (2016): 157-325.
>
> [2] Braun, Gábor, et al. "Conditional gradient methods." arXiv preprint arXiv:2211.14103 (2022).
>
>
>
>
> > `are simulation results important in this area? Would the paper benefit by showing empirical results that prove the performance of the algorithm? I did not see any included.'
>
> We thank the reviewer for this suggestion. We agree that a comprehensive empirical evaluation would be a valuable contribution that could complement our theoretical results.
>
> However, the primary contribution of this work is theoretical. Our main objective is to introduce the first decentralized, projection-free framework for the general class of upper-linearizable functions and to establish its formal performance guarantees through rigorous mathematical proofs. As our claims about regret bounds and complexity are entirely theoretical, we believe the provided proofs serve as the core validation for the paper's contributions.
>
> Nevertheless, we see an empirical investigation of the 10 proposed algorithms as a crucial next step. We have added a note to our conclusion to explicitly highlight a full empirical study as an important direction for future research.
>
> > `the action set is introduced before assumption 2 on page 5, but what is the action set?'
>
> Thank you for catching this. We have revised the word 'action set' here to 'feasible set', and added a footnote mentioning that
> 'In our setting, at each round, every agent $i$ selects a decision from a common feasible set $\mathcal{K}\in\mathbb{R}^d$, which we refer to as the action set. The detailed problem setting is given below in Section~3.3.'

---

> ### Author Response · Authors · 2025-09-23
> **Response to Reviewer 1K9M (2)**
>
> > `were projection-based algorithms considered? Is there a theoretical or empirical result that makes it clear that projection-free algorithms would always be faster?'
>
> To answer directly: yes, projection-based algorithms were considered - the DOBGA algorithm proposed in [4] is indeed projection-based and we have included it in our comparison table, and the same paper also proposed projection-free algorithm Mono-DMFW. Through numerical experiments, [4] shows that Mono-DMFW is much faster than DOBGA in Table 2.
>
> Our choice to develop a projection-free algorithm is not based on an assumption that they are always faster, but on the well-established theoretical and practical advantages they hold for specific, important classes of problems (Chapter 7 in [1], Table 1.1 in [2]).
> The theoretical justification is that the efficiency of each approach depends on the complexity of the feasible set $\mathcal{K}$.
> A projection-based method is faster when the projection operation (a quadratic program) is easy to compute.
> A projection-free method is significantly faster when the projection is computationally prohibitive, but maximizing a linear function (the LOO call) remains efficient.
>
> As we have now clarified in the introduction, a key motivating example is optimization over a nuclear norm ball, where the projection requires a costly full SVD. In such well-documented cases, the projection-free approach is substantially more efficient.
> This focus is consistent with the established literature in decentralized online DR-submodular maximization. Previous works ([3,4,5]) in this area have prioritized the projection-free approach. Notably, the foundational paper by Zhu et al. (2021)[3], which initiated this line of research, states that even for the simpler case of decentralized online convex optimization, "the projection operation can be prohibitive when dealing with high-dimensional data."
> Given that the up-concave and upper-linearizable functions we study are more general than standard convex functions, we believe the motivation to adopt a less prohibitive, projection-free approach is even more compelling. It ensures our framework remains computationally feasible for the challenging, high-dimensional problems that are of practical interest.
>
> We have also added the empirical result from [3] to the second paragraph in Introduction.
>
> [3] Junlong Zhu, Qingtao Wu, Mingchuan Zhang, Ruijuan Zheng, and Keqin Li. Projection-free decentralized online learning for submodular maximization over time-varying networks. Journal of Machine Learning Research, 22(51):1–42, 2021.
>
> [4] Qixin Zhang, Zengde Deng, Xiangru Jian, Zaiyi Chen, Haoyuan Hu, and Yu Yang. Communication-efficient decentralized online continuous dr-submodular maximization. In Proceedings of the 32nd ACM International Conference on Information and Knowledge Management, pp. 3330–3339, 2023.
>
> [5] Yucheng Liao, Yuanyu Wan, Chang Yao, and Mingli Song. Improved projection-free online continuous submodular maximization. arXiv preprint arXiv:2305.18442, 2023.

---

> > ### Comment · Reviewer_1K9M · 2025-11-10
> >
> > I thank the authors for addressing my points and editing the draft accordingly. I don't have further comments.

---

### Review · Reviewer_81GD · 2025-09-09

**Summary Of Contributions:**

I reviewed this paper for its first submission, and felt that there were genuine technical contributions, but the technical presentation was quite lacking. Here I re-use parts of my original review that remain relevant, with updates made according to changes made by the authors in their revised version.

The broad context for this paper is online learning, where the typical "regret" notion is formulated using a sum of loss functions which are not convex, but fall into the category of "weakly up-concave functions". The particular focus of this paper is that of the *decentralized* setting, where there are $N$ separate agents, each of which receives their own (potentially adversarial) loss function at each round, and the ultimate goal is to obtain a small regret, having aggregated over all agents, noting that communication between agents is allowed.

To the best of my understanding, the main results are chiefly technical developments building directly upon existing work, in particular the algorithm design strategy of Pedramfar and Aggarwal (2024) used for "upper-linearizable" functions, originally in the centralized setting. Their results take the form of upper bounds on the (expected) regret associated with a general-purpose algorithm (their Algorithm 1), with different algorithmic settings used for different problem settings (e.g., different types of feedback availability), and they "parameterize" these bounds (see for example $\\theta$ in Table 1) to emphasize how regret bound and communication cost tradeoffs interact.

It should also be noted that another important trait of their approach is that it is "projection free"; the cost that is incurred instead is reflected via the "LOO calls" cost, namely the number of calls to a linear optimization oracle.

**Audience:**

Yes

**Audience Explanation:**

While somewhat technical and terse, there are many readers in de-centralized extensions of general-purpose online learning algorithms, and since the authors are quite methodical in describing what is technically new and of interest here, I believe there is an audience for this paper.

**Broader Impact Concerns:**

Not applicable.

**Claims And Evidence:**

Yes

**Claims Explanation:**

The authors are quite methodical in organizing their exposition of previous literature and describing its relation to the new results obtained in this paper. The overall structure of the paper is good, with a key proof sketch, and highlights of key results summarized in a concise and fairly effective way. I do feel that the technical exposition could be made more accessible, but overall, the authors have addressed the key points I raised in my review for the first version of the paper.

Below are just a couple minor comments.

- In the key regret definition on p.7, $\\hat{\\mathbf{x}}\_{t}^{i}$ appears (note the hat), but this notation is undefined. I know it appears later in Algorithm 1 and is probably just a placeholder, but some this should not be left without mention.

- Regarding the following comment to the original submission, the authors responded to my comment, and there is a brief mention of the role of the two alphas in section 3.2, but I think this needs to be emphasized when defining the regret. *The notion of $\\alpha$-regret is given here without additional comment; does this $\\alpha$-based generality have any meaning in this paper whatsoever? It doesn't appear anywhere else. Furthermore, in the critical definition of upper-linearizable functions, once again "$\\alpha$" appears, but it is not clear if the authors intend this $\\alpha$ to align with the $\\alpha$ that appears in the regret definition. All this is sloppy formulation.*

**Requested Changes:**

Please see the comments raised above. Overall, I think the re-submitted paper addresses the main concerns I had with the original submission.

---

> ### Author Response · Authors · 2025-09-23
> **Response to Reviewer 81GD**
>
> Many thanks to the continued effort from the reviewer to go through our paper and the acknolwedgement of our revision effort. Here are our responses.
>
> > In the key regret definition on p.7, $\hat{\mathbb{x}}_t^i$ appears (note the hat), but this notation is undefined. I know it appears later in Algorithm 1 and is probably just a placeholder, but some this should not be left without mention.
>
> Thank you for catching this. In the revision, in the general case in Problem Formulation in Section 3.3, we have changed the notation $\mathbb{x}_t^i$ to $\hat{\mathbb{x}}_t^i$, to align with our notation for action selected by algorithm in the key regret definition and to remove any confusion. The regret compares optimal action with the action selected by the algorithm. For Algorithm~1, the action selected is $\hat{\mathbb{x}}_t^i$, so that goes into the expression of regret; $x_t^i$ is a placeholder, $\hat{x}_t^i$ is the meaningful thing that is the action.
>
>
> > Regarding the following comment to the original submission, the authors responded to my comment, and there is a brief mention of the role of the two alphas in section 3.2, but I think this needs to be emphasized when defining the regret. The notion of $\alpha$-regret is given here without additional comment; does this $\alpha$-based generality have any meaning in this paper whatsoever? It doesn't appear anywhere else. Furthermore, in the critical definition of upper-linearizable functions, once again $\alpha$ appears, but it is not clear if the authors intend this $\alpha$ to align with the $\alpha$ that appears in the regret definition. All this is sloppy formulation.
>
> Thank you for raising this question.
> In general, we want to minimize the $\alpha$-regret with the highest choice of $\alpha$ possible.
> In concave optimization, the choice of $\alpha=1$ is possible and this reduces to the classical notion of regret.
> However, solving the problem with $\alpha=1$ is, in general, NP-hard.
> So there are two problems.
>
> 1- find the highest $alpha$ where we can achieve sublinear regret; and
>
> 2- minimize the $\alpha$-regret for this choice of $\alpha$.
>
> Given a function class, finding the optimal $\alpha$ is a very difficult mathematical problem.
> If a function class is upper-linearizable with coefficient $\alpha$, then, as shown in [1], it is possible to obtain sublinear $\alpha$-regret in the corresponding offline and online optimization problems.
> Note that this does not say anything about the optimality of such $\alpha$.
>
> To make these points clear we have added a remark after the definition of $\alpha$-regret in page 7:
>
> `As per our earlier discussion in Section 3.1, if we set $\alpha=1$, obtaining a sublinear $\alpha$-regret even the offline centralized version of the problem could be NP-hard.
> Thus, the goal is to find the highest $\alpha$ possible and minimize the $\alpha$-regret for such a choice of $\alpha$.
> \emph{As shown in [1], if a function class is upper-linearizable with constant $\alpha$ (as per Equation 2), then there are algorithms obtaining sub-linear $\alpha$-regret in the corresponding offline (and online) optimization problems.}
> In Cases A.2 and A.3, (in the case $\gamma=1$) the optimal approximation coefficient for the offline problem is known (See [2,3]) and the function classes in A.2 and A.3 are upper-linearizable with these approximation coefficients.
> Moreover, for case A.1, the function class is linearizable with the coefficient $\gamma^2/(1 + c\gamma^2)$ which is the best known approximation coefficient for the corresponding offline optimization problem.
> \footnote{In fact, as mentioned earlier, at least in the case $c=\gamma=1$, this coefficient is conjectured to be optimal. (See [4])}
> Thus, among the results in this work for Cases A.1-A.3, it is only for the case A.1 where a higher approximation coefficient is not yet theoretically ruled out.'
>
>
>
> [1] Mohammad Pedramfar and Vaneet Aggarwal. From linear to linearizable optimization: A novel framework with applications to stationary and non-stationary dr-submodular optimization. Advances in Neural Information Processing Systems, 2024a.
>
> [2] Andrew An Bian, Baharan Mirzasoleiman, Joachim Buhmann, and Andreas Krause. Guaranteed Non-convex Optimization: Submodular Maximization over Continuous Domains. In Proceedings of the 20th International Conference on Artificial Intelligence and Statistics, April 2017b.
>
> [3] Loay Mualem and Moran Feldman. Resolving the approximability of offline and online non-monotone DR-submodular maximization over general convex sets. In Proceedings of The 26th International Conference on Artificial Intelligence and Statistics, April 2023.
>
> [4] Mohammad Pedramfar, Christopher John Quinn, and Vaneet Aggarwal. A unified approach for maximizing continuous DR-submodular functions. In Thirty-seventh Conference on Neural Information Processing Systems, 2023.

---

### Review · Reviewer_Quag · 2025-10-02

**Summary Of Contributions:**

This paper studies decentralized projection-free online convex optimization for DR-submodular functions. Previous work assumes 1-weakly DR-submodular functions with 0 in the feasible set, while this paper extends the analysis to the more general class of upper-linearizable functions, which includes many underexplored settings as special cases. To address this class of problems, the authors extend the centralized counterpart (Pedramfar & Aggarwal, 2024a) to the decentralized setting and provide corresponding theoretical guarantees. They further extend their approach to more general settings, such as the BCO setting and semi-bandit feedback.

**Additional Comments:**

I would like to apologize to the authors and AE for the delay in submitting my review.

**Audience:**

Yes

**Audience Explanation:**

This paper studies decentralized DR-submotular OCO, which I think is a intereting topic and is relvernt to audience in online learning, distributed optimization, and submotular optimization.

**Claims And Evidence:**

Yes

**Claims Explanation:**

This paper is generally well-written, and I especially appreciate that the authors state all notations and claims in a clear and rigorous manner. I also went through the proof of the main theorem and did not find any significant errors. However, I did notice quite a few typos, which I have listed at the end.

**Requested Changes:**

This paper is well-written in general and I find the claims are rigorous. However, there exist lots of typos and some of which I listed below. More specfically:

Page 5, Section 3.1: A set ${K}\in \mathbb{R}^d$ should be  ${K}\subseteq \mathbb{R}^d$ (also happens in footnote 2);

Page 5, Section 3.1: Same paragraph, aff(D) and relint(D): should be  aff(K) and relint(K);

Page 6, Line 1: \forall x\leq y \in K;

Section 3.3, paragraph 3: it should be \hat{x} not x, as \hat{x} is what is played by the learner;

Page 8, paragraph 3, line 2: y' should be \widetilde{y}

Page 8, paragraph 3, line 4: unto->onto

Section 4.1, paragraph 1: We should assume without loss of generality that T mod K =0; in line 2, i should be $i$

Algorithm 1, Step 1: mathbf{A} should be A

Theorem 1: regret R should be \mathcal{R}

Other questions:

In this work, the authors consider both projection-free methods and dealing with broder class of dunctions (i.e., upper-linearizable). While it is a great contribution, I wonder if there is a deeper connection beteen projection-free and upper-linearizable functions? In other words, how do we deal with upper-linearizable functions in projection based setting? Are the oracle in the projection-free setting necessary for dealing with upper-linearizable functions?

The authors considered using one-point feedback to estimte the graident. In classical BCO (e.g., Flexman et al., 2005), people have also consdered using 2-point feedback. Can these methods also applible to this setting (Note that I am not asking for a full extension, a short discussion would be enough)?

What is the paramter $\beta$ in the upper bound in Thoerm 1?

Could the authors define the function $h$ is a more clear way?

Under Table 1, could the authors add more discussion on the comparsion between guruantees of the proposed methods with that of the existing work?

---

> ### Author Response · Authors · 2025-10-07
> **Response to Reviewer Quag**
>
> We thank the reviewer for their thorough reading of our paper. We have corrected the typos they indicated. Below, we address the reviewer's questions and describe the corresponding revisions to our manuscript.
>
> > In this work, the authors consider both projection-free methods and dealing with broder class of dunctions (i.e., upper-linearizable). While it is a great contribution, I wonder if there is a deeper connection beteen projection-free and upper-linearizable functions? In other words, how do we deal with upper-linearizable functions in projection based setting? Are the oracle in the projection-free setting necessary for dealing with upper-linearizable functions?
>
> Thank you for raising this question regarding the relationship between upper-linearizable functions and projection-free methods.
>
> To answer directly: these two concepts are **orthogonal**. "Upper-linearizability" is a property of the function class, while "projection-free" is a property of the optimization algorithm.
>
> **Relationship between the Concepts**: The upper-linearizable property (Equation 2) guarantees that a function's value can be upper-bounded by a linear approximation constructed from a surrogate vector $\mathfrak{g}(f,\mathbf{x})$. This vector, obtained from a "linearizable query oracle," provides a gradient-like direction that is essential for analysis and optimization, regardless of the specific algorithm used to maintain feasibility.
>
> **Optimizing Upper-Linearizable Functions in a Projection-Based Setting**: It is possible to design a projection-based algorithm for this function class. Such an algorithm would operate as follows:
> At each block $m$, each agent $i$ would still query the linearizable oracle to get an estimate of the surrogate vector (suppose we call the aggregate estimate $G^i_m=\sum_{t\in \mathcal{T}_m}o^i_t$).
> The agent would then perform a consensus step and a gradient-ascent-like update: $\mathbf{y}^i\_{m+1}=\sum\_{j\in\mathcal{N}\_i}a\_{ij}\mathbf{x}^j\_m+\eta G^i\_m$.
> Finally, instead of using an infeasible projection oracle (which relies on an LOO), the agent would use a standard projection oracle to ensure the next iterate is feasible: $\mathbf{x}^i\_{m+1}=\mathcal{O}\_{P}(\mathcal{K},\mathbf{y}^i\_{m+1})$. However, we focus on projection-free algorithms due to their lower complexity than projections.
>
> **Necessity of the Oracles**: The linearizable query oracle (which provides $\mathfrak{g}(f,\mathbf{x})$) is fundamental to optimizing this function class as it provides the necessary directional information. However, the linear optimization oracle (LOO) is not inherently required by the function class itself; rather, it is a cornerstone of the projection-free algorithmic paradigm. A possible projection-based algorithm is to simply replace the LOO-based infeasible projection with a projection oracle.
>
> Our choice to pursue a projection-free approach was a deliberate contribution motivated by computational efficiency. As we note in the introduction [1,2], for many complex constraint sets, the LOO is significantly cheaper than a full projection, making projection-free methods more scalable and practical.
> Thank you again for this question, which has helped us clarify the positioning of our work.
>
> [1] Gabor Braun, Alejandro Carderera, Cyrille W Combettes, Hamed Hassani, Amin Karbasi, Aryan Mokhtari,
> and Sebastian Pokutta. Conditional gradient methods. arXiv preprint arXiv:2211.14103, 2022.
>
> [2] Elad Hazan et al. Introduction to online convex optimization. Foundations and Trends® in Optimization, 2
> (3-4):157–325, 2016.
>
>
>
> > Under Table 1, could the authors add more discussion on the comparsion between guruantees of the proposed methods with that of the existing work?
>
> Thank you for this suggestion. Following your advice, we have added the following discussion under Table 1:
>
> 'Notably, all prior works are confined to a very narrow subclass of functions: monotone $1$-weakly up-concave (i.e., DR-submodular) functions over convex sets containing the origin, with approximation ratio of $1-e^{-1}$. In contrast, our framework provides the first guarantees for a much broader range of problems, including general $\gamma$-weakly, non-monotone functions, and optimization over general convex domains. It also introduces a flexible trade-off between regret and communication via the parameter $\theta$. Even when specialized, our results are highly competitive: setting $\theta=\frac{1}{2}$ matches the state-of-the-art projection-free method (DPOBGA), while $\theta=1$ matches the regret of the best projection-based method (DOBGA).'

---

> ### Author Response · Authors · 2025-10-07
> **Response to Reviewer Quag 2**
>
> > The authors considered using one-point feedback to estimte the graident. In classical BCO (e.g., Flexman et al., 2005), people have also consdered using 2-point feedback. Can these methods also applible to this setting (Note that I am not asking for a full extension, a short discussion would be enough)?
>
> Thank you for this valuable suggestion.
> A 2-point gradient estimator, such as the one used in [3,4], provides a better estimate of the gradient. However, it requires access to exact function values. If an exact value oracle is available and multiple queries are possible, a 2-point estimator is indeed superior to a 1-point estimator.
> The crucial point, however, is that using a 2-point gradient estimator requires an exact value oracle, which is often an impractical assumption.
> In the case where an exact value oracle is available, one could use an argument similar to that in [5] to obtain a regret bound of the same order as in the first-order case. In other words, as shown in [3,4] and [5], access to an exact value oracle can lead to a regret bound of the same order as that achieved with a stochastic first-order oracle.
>
> We have added a short discussion (Remark 3) to Section 5.1 of the manuscript after Theorem 3 to clarify this for future readers:
>
> 'Our approach to gradient estimation from zeroth-order feedback relies on the one-point gradient
> estimator (Flaxman et al., 2004). An alternative, common in the bandit optimization literature (Agarwal
> et al., 2010; Shamir, 2017), is to use a two-point estimator, which queries the function at two points (e.g., $f(\mathbf{x}+\delta\mathbf{v})$ and $f(\mathbf{x}-\delta\mathbf{v})$) to construct a finite-difference approximation of the true gradient. While a two-point estimator can provide a better estimate of the gradient, it requires access to an exact value oracle, which is often an impractical assumption. If an exact value oracle is indeed available, then one may use an approach
> similar to Pedramfar & Aggarwal (2024a, Algorithm 7) to develop a counterpart of Algorithm 2 using the
> two-point gradient estimator. Note that such an algorithm will not be bandit, as it requires two queries per timestep. However, we may use arguments similar to Pedramfar & Aggarwal (2024a, Corollary 5) to see that
> such an algorithm has the same order of regret as the first-order algorithm it is based on. In other words, we
> obtain a regret bound with the same order as Theorem 2.'
>
> We have also added a similar short paragraph after Theorem 5 in Section 5.2.2:
>
> 'Similar to Remark 3, if an exact value oracle for the objective function is available, one can develop an
> equivalence of Algorithm 4 using two-point query instead of one-point query, and obtain a regret at the same
> order of Theorem 2.'
>
> [3] Optimal Algorithms for Online Convex Optimization with Multi-Point Bandit Feedback. - Agarwal, Dekel, Xiao - 2010
>
> [4] An Optimal Algorithm for Bandit and Zero-Order Convex Optimization with Two-Point Feedback - Shamir - 2017
>
> [5] Mohammad Pedramfar and Vaneet Aggarwal. From linear to linearizable optimization: A novel framework with applications to stationary and non-stationary dr-submodular optimization. Advances in Neural Information Processing Systems, 2024a.

---

> ### Author Response · Authors · 2025-10-07
> **Response to Reviewer Quag 3**
>
> > What is the paramter $\beta$ in the upper bound in Thoerm 1?
>
> Thank you for pointing this out. The parameter $\beta$ is a positive constant from the definition of the upper-linearizable function class, which we introduce in Section 3.2, Equation (2). Specifically, Assumption 3 states that 'All objective functions $f_{t,i}\in\mathcal{F}:\mathcal{K}\to \mathbb{R}$ are $M_1$-Lipschitz continuous, differentiable, and upper-linearizable with $\alpha, \beta,\frak{g}$ and $h$ as defined in Equation (2).'
>
> > Could the authors define the function $h$ is a more clear way?
>
> We note that the definition of $h$ is part of the definition of upper-linearizability and unrelated to decentralization, or projection-free/projection-based optimization.
> If the notion of upper-linearizabibility was defined without using $h$ (i.e., with $h$ being identity), the notion would have been much less expressive.
> To clarify the role of $h$, we can provide an alternative perspective on upper-linearizability by first re-examining concavity.
>
> We start with a definition.
> Let $\mathcal{F}$ be a class of functions over a convex set $\mathcal{K} \subseteq \mathbb{R}^d$ and let $L$ be a functional such that, for any $f \in \mathcal{F}$ and $x \in \mathcal{K}$, $L_{f, x}$ is an affine map over $\mathbb{R}^d$. Let us call such a functional $L$ a linear assignment for $\mathcal{F}$.
> The following lemma follows from the definition of concavity.
>
> Lemmma:
> A function class $\mathcal{F}$ consists only of concave functions if and only if it has a linear assignment $L$ such that, for all $f \in \mathcal{F}$ and $x, y \in \mathcal{K}$, we have
> $L_{f, x}(x) = f(x)$
> and
> $L_{f, x}(y) \geq f(y)$.
>
> Basically, the above lemma relates the notion of concavity with the idea of the tangent plane being an upper bound for the function.
> Now we phrase the definition of upper-linearizability in a way that is similar to the above lemma.
> We can also see clearly why such function classes are called "upper-linearizable".
>
> A function class $\mathcal{F}$ is called upper-linearizable if and only if it has a linear assignment $L$ such that, for all $f \in \mathcal{F}$ and $x, y \in \mathcal{K}$, we have
> $L_{f, x}(x) = \frac{1}{\alpha}f(h(x))$
> and
> $L_{f, x}(y) \geq f(y)$,
> for some function $h : \mathcal{K} \to \mathcal{K}$ and some $\alpha \in (0, 1]$.
>
> Given this equivalent definition, it is clear that droping $h$ (i.e., assuming $h(x)=x$) significantly reduces the expressive power of the notion of linearizability.
>
> Since this is tangential to the main contribution of this work, we have added a separate **Appendix A** in the revision to further clarify the connection between this concept and concavity, which should illusminate the role of $h$. In Section 3.2 in the main text, we have added a brief discussion that refers to Appendix A for a detailed discussion:
>
> 'We note that the function h allows us to consider more general functions. For example, $h(\cdot)$ takes identity
> function for case B.1 and case B.2, while $h(\mathbf{x})=\frac{\mathbf{x}+\bar{\mathbf{x}}}{2}$ for some constant $\bar{\mathbf{x}}\in\mathcal{K}$ for case B.3. A detailed discussion on the role of the function $h$ is presented in Appendix A.'

---

> > ### Comment · Reviewer_Quag · 2025-10-07
> > **Response**
> >
> > I would like to thank the authors for the detailed response and I do not have further questions.

---

### Decision · Action_Editor_UjAj · 2025-11-22

**Recommendation:** Accept as is

**Audience:**

Yes

**Audience Explanation:**

There will be some, but limited interest in the paper by the community. However, it is worth bringing the results to the public.

**Claims And Evidence:**

Yes

**Claims Explanation:**

This paper extends its centralized counterpart (Pedramfar & Aggarwal, 2024a) to the decentralized setting. Efforts are also made to incorporate classical techniques from the BCO/semi-bandit literature. All reviewers reach a consensus that the main claims made by the manuscript are likely correct.